# Incorporating Uncertainty in Machine Learning Models to Improve Early Detection of Flavescence Dorée: A Demonstration of Applicability [note 1]

**DOI:** 10.3390/s25247493

**Published:** 2025-12-09

**Authors:** Cristina Nuzzi, Erica Saldi, Ilaria Negri, Simone Pasinetti

**Affiliations:** 1Department of Mechanical and Industrial Engineering, University of Brescia, Via Branze 38, 25123 Brescia, Italy; simone.pasinetti@unibs.it; 2Department of Sustainable Crop Productions, Catholic University of the Sacred Heart, Via E. Parmense 84, 29122 Piacenza, Italy; ericasaldi96@gmail.com (E.S.); ilaria.negri@unicatt.it (I.N.)

**Keywords:** Flavescence dorée, hyperspectral imaging, machine learning, measurement science, plant disease detection, uncertainty

## Abstract

Early detection of Flavescence dorée leaf symptoms remains an open question for the research community. This work tries to fill this gap by proposing a methodology exploiting per-pixel data obtained from hyperspectral imaging to produce features suitable for machine learning training. However, since asymptomatic samples are similar to healthy samples, we propose “uncertainty-aware” models that address the probability of the samples being similar, thus producing, as output, an “unclassified” category when the uncertainty between multiple classes is too high. The original dataset of leaves hypercubes was collected in a field of Pinot Noir in northern Italy during 2023 and 2024, for a total of 201 hypercubes equally divided into three classes (“healthy”, “asymptomatic”, “diseased”). Feature predictors were 4 for each of the 10 vegetation indices (population quartiles 25-50-75 and population’s mean), for a total of 40 predictors in total per leaf. Due to the low number of samples, it was not possible to estimate the uncertainty of the input data reliably. Thus, we adopted a double Monte Carlo procedure: First, we generated 30,000 synthetic hypercubes, thus computing the per class variance of each feature predictor. Second, we used this variance (serving as uncertainty of the input data) to generate 60,000 new predictors starting from the data in the test dataset. The trained models were therefore tested on these new data, and their predictions were further examined by a Bayesian test for validation purposes. It is highlighted that the proposed method notably improves recognition of “asymptomatic” samples with respect to the original models. The best model structure is the Decision Tree, achieving a prediction accuracy for “asymptomatic” samples of 75.7% against the original 49.3% for the Ensemble of Bagged Decision Trees (ML4) and of 44.6% against the original 13.2% for the Coarse Decision Tree (ML1).

## 1. Introduction

The phytoplasma of Flavescence dorée (FD) is transmitted to the plants through the saliva of infected insects, mainly leafhoppers of the species *Scaphoideus titanus* that feed on the plant’s leaves [1]. The insect is native to North America and was first observed in France in the 1960s [2], and it is now spread throughout Europe [3]. Since 2024, the presence of *Scaphoideus titanus* has also been confirmed in Germany, in vineyards bordering France and Switzerland [4]. The known symptoms of FD are described in the European Food and Safety Authority “Pest Survey Card” on the phytoplasma [3]. They appear gradually over the summer, with a peak at the end of August. Infected plants appear ill and droop due to a lack of lignification, and inflorescences and berries quickly die over the season. Leaf symptoms are the most evident and include a change in color and texture, rolling downwards gradually over time; in the case of red grapes, the leaves become red, and they become yellow for white grapes. Despite being a quarantine disease in Europe [5], the presence of infected vineyards that do not undergo mandatory treatments (thus becoming hot spots for the disease) is a further threat to nearby vineyards. Most control strategies focus primarily on the insect. Treatments include traditional approaches such as chemical pesticides or novel methods like natural products [6] or microorganisms with bio-control properties [7]. In addition, yellow sticky traps are commonly adopted to monitor the number of insects in the vineyard, a useful practice to plan extra treatments in the case of severe infestations.

The infection typically happens during winter, so the first symptoms can be spotted in spring when it is too late to limit the contagion to nearby plants. This is why early detection of the disease is a necessity for vine-growers. However, this is not an easy task since FD symptoms may be confused with those of other diseases (e.g., *Bois noir*) or conditions (e.g., water or redrednutritional stress, senescence). Currently, the presence or absence of FD can be only certified by a Polymerase Chain Reaction (PCR) analysis on leaf samples [8], which is an expensive analysis requiring dedicated laboratories and trained personnel. Unfortunately, this method is effective only if the symptoms are severe, because the molecular analysis is not sensitive to low quantities of the disease in the leaf sample, making it useless for early detection on asymptomatic samples. According to [9,10,11,12,13], most physiological changes can be detected in the visible spectral range and in the range of 1600–2200 nm, which is related to water absorption and phenolic compounds accumulation effects.

Other options that are quickly gaining popularity involve contactless visual measurements such as combinations of visible and near-infrared (NIR) sensors as in [14,15,16], multispectral imaging (MSI), and hyperspectral imaging (HSI). The main advantage of these two methods is that the spectral information is acquired for each pixel of the leaf sample at once, thus keeping the spatial information of the leaf’s visual details. In MSI, cameras are equipped with filters for specific spectral bands to capture images at different wavelengths during movement. However, the spectral information produced is less dense compared to HSI, since the sample’s reflectance in the wavelengths belonging to the spectral band is fused, while HSI devices will produce outputs for each wavelength (according to their spectral resolution). Multispectral cameras can be found on the market according to the application needs or can be built and customized by combining imaging devices and filters, as in [17]. Often, MSI is coupled with vegetation indices (VIs), providing a set of indicators to the farmer describing the overall health of the field [18,19,20]. These works demonstrated that the most informative spectral bands to accurately detect FD infection differ according to the vine variety; thus, the best VIs for the task change as well. These works also highlighted that MSI is not sufficiently accurate to detect subtle changes in the spectral response of leaves, making it unsuitable for the early detection of FD symptoms. Moreover, remote sensing solutions imply a low image resolution since the camera is distant from the plant [21,22]. In contrast, HSI is the most accurate option that allows the acquisition of dense hypercubes in a specific sensing range with high spectral resolution and is specifically suited for inspecting and analyzing biological samples, as demonstrated by [23]. In particular, HSI allows the development of more refined detection algorithms based on the combination of subtle changes in the spectral response of the sample and spatial information such as the sample’s size, surface texture, geometric patterns, and their location on the sample’s surface. This allows the adoption of traditional computer vision approaches [24] as well as methods belonging to the broad field of artificial intelligence, such as machine learning (ML) and deep learning (DL) models and algorithms, extensively adopted in the field of agriculture and especially for applications related to vineyard monitoring [25,26]. It is worth noting that, at the present time, the research community focuses mainly on the detection of FD by classifying samples into two classes (healthy or diseased), without considering asymptomatic samples [27,28,29]. In our previous works [30,31], HSI was coupled with ML to classify samples into three classes (healthy, asymptomatic, diseased) according to the VIs calculated from the average spectrum of each leaf sample. Currently, no works address classification uncertainty for this specific research problem.

The present article is a technically extended version of our previous works [30,31] with the following improvements: (1) the samples collected for each class are now balanced, and the dataset is doubled in size; (2) the processing algorithm developed now extracts per-pixel features instead of per-sample features, resulting in more refined predictors that characterize a sample; (3) we present a novel methodology based on Monte Carlo augmenting that integrates uncertainty with the predictions of ML classification models, resulting in “uncertainty-aware” ML models that behave better when a certain sample could potentially belong to more than one class. It is important to stress that the goal of the present article is to demonstrate the validity of the approach for FD detection also in asymptomatic samples, exploiting the concept of uncertainty; as a result, the focus is not on the ML model adopted but rather on the analysis and validation procedure, which can be applied to a plethora of ML models.

## 2. Materials and Methods

### 2.1. Materials

The hyperspectral camera adopted was the same as in [30,31], which is the HERA VIS-NIR camera (NIREOS S.r.l., Milan, Italy) with a spectral range of 400–1000 nm and a spectral resolution of 5 nm (120 bands). As stated by the manufacturer, since the camera’s acquisition technology is based on Fourier Transform, only intensity calibration was necessary during measurement (performed using a reference teflon sample as suggested by the manufacturer’s guidelines). Geometric calibration was already provided as well in the proprietary software used for the acquisitions. The camera was managed by a laptop computer equipped with Windows 11 Pro, CPU Intel i7 2.80 GHz, and 32 GB RAM. The processing software and the ML model were developed using MATLAB 2024b (The Mathworks Inc., Natick, MA, USA) on a laptop equipped with an Intel i5 1.60 GHz CPU, 8 GB of RAM, and 500 GB of storage space. The generation of synthetic samples detailed in Section 2.5.1 was conducted in parallel on the former machine and on two workstations equipped with an i7 2.50 GHz CPU with 32 cores, 32 GB of RAM, and 2 TB of storage space.

### 2.2. Data Acquisition Procedure

Leaf samples were collected in two campaigns from the same field of Pinot Noir located in Pometo (Colli Verdi, Pavia, Italy): one in 2023 and another in 2024. In both cases, the collection of the samples was conducted with the aid of an expert agronomist trained to recognize FD symptoms. The data collected from both campaigns is balanced per-class and is detailed in Table 1, showing the number of samples per class.

Campaign 1 (2023): Leaf samples were collected from plants suspected of being positive to FD and marked with colored strips by the expert agronomist found in all the vineyard without limiting the search to a single row. Healthy samples were collected from Pinot Noir plants cultivated in a protected environment. Collection days were the 14th of July, the 6th of September, and the 18th of September. Details about this campaign and the relative data analysis can be found in our previous work [30].Campaign 2 (2024): Leaf samples were collected from plants belonging to just one row of the vineyard. The selected row did not contain plants that were affected by FD during 2023. Collection days were the 31st of May, the 1st of July, the 6th of August, and the 20th of September. Details about this campaign and the relative data analysis can be found in our previous work [31].

The hypercube of each leaf sample was acquired approximately 15 min after all the leaves were taken from the field during the specific collection day. This time is well below the dehydration time limit as stated in [32] thanks to the moderate respiration rate of vine leaves; nonetheless, after taking a leaf from the plant, it was immediately placed in water to preserve it and avoid dehydration. Each leaf was then cleaned and dried before acquiring its hypercube. The acquisitions were conducted indoors in a facility nearby equipped with a window from 10 AM to 12 AM. This was necessary because the field was lacking electric power to run the equipment. The leaves were placed on a white paper sheet positioned on the ground. The camera-object distance was set to 1 m, ensuring that the leaf was always in focus. To this distance corresponds a Field of View (FoV) of 271×217 mm. Only natural light was coming from the outside, and there were not overexposed areas on the leaves’ surface. The acquisition and saving of a single hypercube lasts on average 10 s. The hypercubes were saved on disk in binary format and elaborated afterward to compute the normalized reflectance spectrum of each pixel in the hypercube. Normalized reflectance allows the minimization of non-biological variations of reflectance due to factors such as leaf tilting, bending, and camera-specific parameters. Normalization was conducted in two steps. First, using the camera’s proprietary software, each leaf’s hypercube was normalized using the data of the hypercube of a reference Teflon sample acquired before the acquisition of the samples. Then, the reflectance values in the resulting hypercube are divided by the mean reflectance across 875–925 nm bands. As stated by the protocols in [33,34], these bands exhibit high reflectance intensities regardless of the leaf’s species and environmental stresses. The normalized hypercubes were then converted into MATLAB format, and each of them required around 1.5 GB of space. Figure 1 shows the per-band average and standard deviation of the normalized reflectance of leaves belonging to each class, computed over the 67 samples per class.

After the hypercubes acquisition, all the leaf samples were analyzed by an external laboratory to detect the presence or absence of FD following Pellettier’s grapevine extraction protocol for phytoplasma detection [8]. The leaves were always collected from the sample plants selected for the experiment during the corresponding campaigns. Therefore, samples that resulted negative to FD at the start of the season but positive at the end were labeled as “asymptomatic” samples. For example, considering three samples taken from the same plant, the ones collected during the first and second acquisition dates were labeled as “asymptomatic”, while the one collected on the third date was labeled “diseased”. Examples of leaf samples per class are shown in Figure 2. For the ML model training and testing, the data shown in Table 1 was divided into two sub-datasets: the training dataset ($D_{\mathrm{train}}$), containing the 70% of the data, and the test dataset ($D_{\mathrm{test}}$), containing the remaining 30% of the data. It is worth noting that class balancing was conducted by manually choosing the maximum number of samples that were labeled “asymptomatic” after PCR analysis (67) and thus selecting the same number of samples for “diseased” and “healthy” categories, chosen among all the collected and PCR-tested samples.

### 2.3. Pre-Processing

The pre-processing procedure shown in Figure 3 was applied to each hypercube in the datasets (for a total of i=1…201 hypercubes) to prepare the data for ML analysis. The steps are described in detail in the following sections.

#### 2.3.1. Selection of Pixels Belonging to the Leaf

As a first step, the original hypercubes of size 1024×1080 px ×120 spectral bands were resized to 640×480 px ×120 spectral bands due to the heavy computational requirements needed to process them during the synthetic data generation described in Section 2.5, obtaining $H_{i}$. To avoid interpolation issues during resizing, each layer of the hypercube (from 1 to 120) was resized to the target size of 640×480 px independently using bicubic interpolation. To assess the validity of the resizing approach and ensure no interpolation distortions were accidentally added to the data, differences between the original mean leaf spectra and the resized mean leaf spectra were computed per sample, resulting in a maximum difference of $2.5\times 10^{-3}\pm 0.8\times 10^{-3}$ (dimensionless) in the infrared region (which is primarily affected by light scattering effects). After resizing, the color image of the hypercube $C_{i}$ is reconstructed by considering the mean of the 3 color channels (red, green, and blue), thus taking into account light variations and reducing color alterations. The spectral bands used to average the channels were: 581–667 nm for red, 510–576 nm for green, and 481–496 nm for blue.

To remove the background pixels and only keep those belonging to the leaf, the easiest way is to apply a binary mask on the hypercube to delete unwanted pixels. The masks $M_{i}$ were computed using the open-source segmentation platform Roboflow [35] that leverages AI to generate polygonal contours of items. In the online platform, polygonal contours of the leaves were generated by manually clicking on the leaf, easily detected by the segmentation model adopted in the platform [36] since all leaves were positioned over a white background, maximizing contrast. The whole process was supervised by a user, and all the eventual mistakes in the contour definition were manually corrected. The output is a file containing the points forming the polygonal contour of each leaf. The contour file was then converted to a binary mask $M_{i}$ in MATLAB afterward using a custom function that creates the mask image from the polygonal contour file produced by Roboflow. For examples of color images and binary masks, the reader is referred to our previous work [30]. Finally, the mask is applied to the hypercube for each spectral band λ=1…120, one at a time, thus extracting only the pixels belonging to the leaf. To do so, data corresponding to spectral band λ is extracted from $H_{i}$ and rolled, obtaining $H_{i,\lambda }$. The mask $M_{i}$ is rolled as well, and then, by multiplying the two, only pixels corresponding to logical values set at 1 in $M_{i}$ are retained in the new $T_{i,\lambda }$. This procedure is repeated for all spectral bands λ, producing a matrix $T_{i}$ of size $N_{px}\times 120$, where $N_{px}$ equals the number of pixels belonging to leaf *i* and 120 is the number of acquired spectral bands. Please note that according to the leaf analyzed, this value changes, ranging from 100,000 px for small leaves to 250,000 px for large leaves. This step is represented by the yellow block in Figure 3.

#### 2.3.2. Features Computation

From the pre-processed leaves $T_{i}$, a total of 10 vegetation indices (VIs) were selected from the literature to describe the health status of the leaf. VIs are designed to summarize the leaf’s reflectance spectrum according to specific areas of interest, e.g., the green, red, or near-infrared spectral regions, and are commonly used by the research community as indicators that facilitate the detection of stress in plants. The mathematical definitions and indices descriptions of the VIs used in this work can be found in Table 2. All VIs are by definition unitless. The impact of natural illumination variability was kept under control during the experiment. However, please note that light scattering effects are mainly affecting wavelengths higher than 800 nm, which were not used by the VIs selected for this work.

All $T_{i}$ contain the spectrum over 120 bands of each pixel of leaf *i*; hence, it is possible to compute per-pixel VIs instead of per-leaf (as performed in [30]), producing a population of values for each feature. Since some pixels may be saturated due to light scattering and reflections, each $T_{i}$ is filtered to remove unwanted outliers. The data of each VI is checked for outliers independently (e.g., by scanning the columns of $T_{i}$). To do this, we used MATLAB’s “isoutlier” function that detects outliers as those data points with a value higher than $Q_{3}+1.5\cdot IQR$ or lower than $Q_{1}-1.5\cdot IQR$ (where $Q_{3}$ and $Q_{1}$ represent the third and first quartiles of the data). This step produces ${\hat{T}}_{i}$ of size $N_{px}\times 10$, with i=1…201.

Since to each VI corresponds a population of $N_{px}$ values, the features selected for ML analysis were computed to reflect the population of VI values of each leaf, corresponding to: (1–3) the three quartile values of the population ($VI_{Q25}$, $VI_{Q50}$, $VI_{Q75}$), and (4) the population’s mean $VI_{m}$. Therefore, for each ${\hat{T}}_{i}$ (e.g., for each leaf), a set of 4 features ×10 VIs =40 ML predictors is produced. The final sizes of the datasets used for ML training and testing is 141×40 for $D_{\mathrm{train}}$ and 60×40 for $D_{\mathrm{test}}$. Data belonging to $D_{\mathrm{train}}$ is then standardized using the z-score method. The mean and standard deviation values resulting from this operation are used to standardize the data in $D_{\mathrm{test}}$ afterward. This final step is common practice when working with features for ML that may have quite different ranges. Z-score standardization was applied globally, without keeping classes separate.

### 2.4. Machine Learning Models Tested

The goal of this study is not to provide a comprehensive comparison among all the existing ML models, but instead to demonstrate the applicability of the proposed method that incorporates uncertainty into existing ML architectures to improve the detection of FD in asymptomatic samples. As a result, we selected only a subset of 5 standard off-the-shelf models among the ones available in MATLAB. In particular, only model architectures that produced a positive prediction outcome related to probabilistic per-class confidence scores (summing to 1) were considered, and only one model per category was included for comparison. It is worth noting that Support Vector Machines and Logistic Regressors produce outputs related to their model’s structures (e.g., support vectors for SVMs) that are then remapped into a class [41,42] instead of confidence scores; hence, they were not considered in this work to avoid issues with the aggregation of probabilities described in Section 2.5.2. The models tested were:ML1: a Coarse Decision Tree with 4 maximum number of splits (criterion: Gini’s diversity index) [42,43].ML2: a Naive Bayes model with Gaussian distribution [44].ML3: a k-Nearest Neighbor (k-NN) model with the number of neighbors equal to 10 and distance metric set to Minkowski cubic distance [45].ML4: an Ensemble of Bagged Decision Trees model with maximum number of splits equal to 140 [43,46]. Ensembling is a method to reduce variance in noisy datasets by training multiple instances of the same model architecture in parallel. Among ensembling methods, bagging is a method in which a random sample of the training set is selected and replaced each time a new instance is trained; thus, the final model will be the aggregation of all the models trained in parallel, and their performance will be averaged. In our case, we set the number of parallel models equal to 30.ML5: a Wide Neural Network with 1 fully connected layer, a first layer size of 100 neurons (with ReLU activation function), and a maximum of 1000 iterations [47].

The inputs of all ML models were the 40 features computed during the pre-processing procedure in Section 2.3. Cross-validation was conducted using 5 folds on $D_{\mathrm{train}}$ in all cases.

### 2.5. Uncertainty-Aware Models

Ideally, when developing a measurement pipeline, the uncertainty associated with the final measurement outcome should be quantified as well. This information is crucial to understanding the applicability and robustness of the measurement pipeline. However, it is not easy to quantify metrological uncertainty in classification models (output is a category) compared to regression models (output is a numerical value).

Measurement uncertainty affects any input data and should be propagated inside the ML model, as the approach described in [48,49] does. However, this is feasible only if the input data uncertainty can be computed and the ML model’s structure allows for propagating it. Therefore, we have to address two issues: (1) the modeling of the input data uncertainty considering the low number of input data contained in our dataset (not enough to properly describe a complex phenomena such as FD), and (2) the generalization of the approach so that it can be applied to every ML model that outputs a confidence score. The procedure described in this Section addresses both issues and is summarized as follows (see Table 3 for a list of relevant parameters used in the mathematical notation):Generation of synthetic variability: To address the issue of having a low number of samples in our dataset, the idea is to generate new samples using Monte Carlo generation starting from the s=60 samples in $D_{\mathrm{test}}$ since they were not seen by the trained ML models. Following the procedure in Section 2.5.1, we generated “synthetic” leaf hypercube samples starting from the original hypercubes in $D_{\mathrm{test}}$ by substituting the spectra contained in each pixel of the original hypercubes with a new one following a Monte Carlo procedure. To do so, we exploited the per-band spectral variability computed by acquiring 50 times the hypercube of a new leaf sample not included in any dataset collected specifically for this test. For each hypercube in $D_{\mathrm{test}}$, we generated M=500 synthetic hypercubes, thus keeping the processing times reasonable considering the high number of pixels contained in them ($N_{px}$ > 100,000 px). We then apply the pre-processing procedure described in Section 2.3.2 to the synthetic hypercubes, obtaining f=40 feature predictors for each synthetic sample. By comparing the variability of the original predictors in $D_{\mathrm{test}}$ divided per class ($\sigma_{1,f,c}$) with the variability of the synthetic data ($\sigma_{2,f,c}$), we can determine if the synthetic data properly models the original phenomena. The resulting $\sigma_{2,f,c}$ is then used as the input data uncertainty in the following step.Classification uncertainty: Following the procedure described in Section 2.5.2, we used the variability $\sigma_{2,f,c}$ to generate new synthetic samples starting from the data in $D_{\mathrm{test}}$ by means of a second Monte Carlo procedure. This allows the generation of feature predictors directly rather than generating a hypercube, which is a faster process in comparison and also ensures we avoid carrying over the uncertainty related to the curve generation procedure used in Section 2.5.1. For each sample *s* in $D_{\mathrm{test}}$ a total of G=1000 new samples were generated this way, producing the new dataset $D_{\mathrm{gaussian}}$ with n=s×G = 60,000 rows. To obtain the uncertainty-aware models, we followed the idea expressed in [48,49] describing a statistical approach to aggregate the traditional confidence scores returned by the m=5 models (namely $P_{m}$), representing the probability of the input sample belonging to each class on which the model was trained, with another set of probabilities (namely $Z_{m}$, also referred to as population uncertainty) representing how likely it is that a certain test sample belongs to a specific class according to whether the features of the sample fall inside the distribution of the training population features or not. Please remember that the prediction vector $p_{n}$ returned by each model for each test sample *n* to be inferred contains the probability from 0 to 1 of the *n*th sample of being of class c=1…3. The sum of the values in all rows of $P_{m}$ (e.g., $p_{n}$) is equal to 1. The predictions can then be arranged in a confusion matrix, which, according to [30], can be used as the starting point to apply a Bayesian test and assess the model’s robustness, as demonstrated in Section 2.5.3.

#### 2.5.1. Generation of Synthetic Variability

To generate the new synthetic per-pixel spectra of each sample, we first have to calculate the variability of the original predictors in $D_{\mathrm{test}}$. This is also expressed as the variability of the *f* predictors of the samples belonging to class *c* (remember that for each class we only have i=20 samples). This variability is computed as the standard deviation $\sigma_{1,f,c}$ per-predictor, considering the samples belonging to each class. It is worth stressing that, as described in Section 2.3.2, the original 10 VIs are computed for each pixel of the leaf sample. Hence, the *f* feature predictors should represent the distribution of the VIs inside the leaf. However, this is a spatial distribution, since it is closely related to the pixels and their locations in the leaf. Instead, $\sigma_{1,f,c}$ represents the variability of predictors among classes; in other words, the distribution of values among the population of samples considered.

To compute the synthetic values, we first analyzed the spectral variability of the samples by conducting repeated measurements of a single healthy leaf sample specifically collected for this test and not included in any dataset. The sample’s hypercube was acquired j=50 times, each time moving it inside the field of view to slightly change the incidence of the light rays on its surface with respect to the camera. This resulted in *j* hypercubes $L_{j}$, each acquired at time intervals of 1 min between each other. Pre-processing steps described in Section 2.3.1 were applied to all $L_{j}$ hypercubes, obtaining $I_{j}$ processed matrices. Each $I_{j}$ contains the variability of the sample. To obtain also the repeatability of the measurement plus the variability of the sample, all processed matrices $I_{j}$ were merged into one, obtaining $S_{1:50}$. Therefore, from $S_{1:50}$ it is possible to calculate the mean $\mu_{\lambda }$ and standard deviation $\sigma_{\lambda }$ per spectral band λ. This results in the specific spectral band variability vector $V_{\lambda }$ of size 1×λ=1×120. This procedure is graphically depicted in the first block in Figure 4.

Now, we can apply a Monte Carlo generation of synthetic samples following [50]. This can be thought of as a method to assess how much the feature predictors computed from the same leaf may change if the measurement is repeated. A leaf sample is represented as a matrix $H_{s}$ (s=1…60 samples in $D_{\mathrm{test}}$) with $N_{px}$ rows and λ columns corresponding to the spectral bands in the range 400–1000 nm. The corresponding synthetic leaf sample $S_{s,M}$ (with M=1…500 Monte Carlo simulations) is composed of $N_{px}$ synthetic spectra. Please note that the number of Monte Carlo simulations *M* is justified by the complexity of the simulations, and it is acceptable as stated in [50]-Section 7.2. Each synthetic spectrum is generated according to the following procedure, graphically depicted in the second block of Figure 4:The spectrum of a single pixel $S_{px}$ is analyzed to find the location of all the peaks and valleys of the signal, described as tuples of $\left(\lambda ,y_{p}\right)$, where λ is the data point wavelength corresponding to the location of the peak/valley and $y_{p}$ is the corresponding normalized reflectance value.For each peak/valley $\left(\lambda ,y_{p}\right)$, the custom function generates a new point $\left(\lambda ,{\hat{y}}_{p}\right)$. ${\hat{y}}_{p}$ is obtained by drawing from a normal distribution with mean equal to $y_{p}$ and standard deviation equal to $\sigma_{\lambda }$.A new curve is generated using a cubic spline that interpolates all the new points $\left(\lambda ,{\hat{y}}_{p}\right)$. This curve is the synthetic spectrum of the pixel, ${\hat{S}}_{px}$.

After generating the s×M = 30,000 synthetic leaves $S_{s,M}$, their VIs values are filtered for outliers and standardized using the mean and standard deviation of $D_{\mathrm{train}}$ as described in Section 2.3.2. Then, the 40 features predictors are computed for each $S_{s,M}$, finally obtaining a synthetic dataset $D_{\mathrm{synth}}$ containing 10,000 synthetic samples per class and their corresponding *f* features (overall size of $D_{\mathrm{synth}}$ is 30,000 rows × 40 features). Their variability $\sigma_{2,f,c}$ is obtained by computing the standard deviation of the *f* predictors calculated from the 10,000 synthetic leaves belonging to class *c* (see last block of Figure 4).

#### 2.5.2. Classification Uncertainty

This step aims to evaluate the uncertainty of the classification models. To do so, we first need to generate new synthetic samples. It is worth noting that the samples in $D_{\mathrm{synth}}$ produced during the experimentation in Section 2.5.1 cannot be used because they would carry over their uncertainty (corresponding to the uncertainty of the Monte Carlo curve generation, which may be different if another method is used instead of the one presented). As a result, we need to generate new samples using a different approach.

Considering the data in $D_{\mathrm{test}}$ (which is a matrix of size s×f=60×40), we can separate the data according to the ground-truth class, obtaining c=3 sub-matrices $D_{\mathrm{test},c}$ each of size i×f=20×40. For simplicity, a single element of these matrices will now be called $d_{c,i,f}$. The idea is to generate new *G* data from each original test sample using a Monte Carlo Gaussian sampling. The sampling is conducted considering the predictor value stored in $d_{c,i,f}$ as the mean μ and as standard deviation σ the corresponding $\sigma_{2,f,c}$ obtained from the procedure described in Section 2.5.1. To keep the computational time reasonable, we chose G=1000. This produced a new dataset $D_{\mathrm{Gaussian}}$ of size n×f=60,000×40 (20,000 samples per class), where n=s×G=60×1000. The data in $D_{\mathrm{Gaussian}}$ is fed into the trained models described in Section 2.4, producing a set of *p* predictions (*p* is a vector of size 1×c) containing the confidence scores of each class for each data sample (as introduced at the start of Section 2.5). This produces matrix $P_{m}$ for each model *m*, of size n×c.

Referring to publications [48,49], we can obtain an uncertainty-aware model *m* by aggregating the probabilities $P_{m}$ obtained from the model with another set of probabilities named $Z_{m}$. Data in $P_{m}$ represents the confidence scores of the model, and their sum is equal to 1. Instead, the probabilities contained in $Z_{m}$ are designed to consider how likely it is that a new sample of the test dataset belongs to the population of the training dataset; hence, their sum is not equal to 1. To express this information, the z-test operator [51] is adopted. The z-test is a statistical operator that outputs a probability representing whether a certain sample being tested belongs to a population with a known mean and standard deviation. Therefore, since we are considering a total of 3 classes with different distributions, the data in $D_{\mathrm{train}}$ was divided into three sub-matrices (one per class), from which we computed the mean $\mu_{c,f}$ and standard deviation $\sigma_{c,f}$ of the predictors that were used for the z-test operation. The features of the data in $D_{\mathrm{Gaussian}}$ were then z-tested using $\mu_{c,f}$ and $\sigma_{c,f}$. This results in a matrix of size f×c for each sample *n* in $D_{\mathrm{Gaussian}}$. We will refer to the 3D matrix of size 40×3×60,000 as ${\mathrm{Ztest}}_{m}$. This step is graphically shown in the first block of Figure 5.

The data in ${\mathrm{Ztest}}_{m}$ was then weighted by feature importance using the formulation from [48,49]. The models weigh the features according to their structure; hence, we have different vectors $w_{m}$ of size f×1=40×1 for each model considered. Weights inside $w_{m}$ were obtained by calculating the Shapley importance [52,53] on the training dataset, which is the state-of-the-art method to calculate feature importance of a generic ML model. Referring to the second block in Figure 5 for a graphical representation of the calculations, the following formula is applied to compute the new values $z_{n,c,m}$:(1)$$z_{n,c,m}=\frac{\prod_{f=1}^{F}{\left(\frac{p_{f,c,n,m}}{1-p_{f,c,n,m}}\right)}^{w_{f,m}}}{1+\prod_{f=1}^{F}{\left(\frac{p_{f,c,n,m}}{1-p_{f,c,n,m}}\right)}^{w_{f,m}}}$$

This produces a matrix $Z_{m}$ of size n×c=60,000×3 of weighted aggregated probabilities. Therefore, for a certain model *m*, we obtain a matrix containing the model’s prediction, $P_{m}$, and a matrix containing the weighted z-scores, $Z_{m}$. To get the final uncertainty-aware model, the probabilities in both matrices $P_{m}$ and $Z_{m}$ should be combined using Equation (Equation 1) once again, modified as follows:(2)$$u_{n,c,m}=\frac{\prod_{i=1}^{V}{\left(\frac{p_{n,c,i}}{1-p_{n,c,i}}\right)}^{1/V}}{1+\prod_{i=1}^{V}{\left(\frac{p_{n,c,i}}{1-p_{n,c,i}}\right)}^{1/V}}$$
where *V* is equal to the number of predictions to aggregate (in our case, V=2 since we want to aggregate predictions contained in both $P_{m}$ and $Z_{m}$), *c* indicates the class, and *n* refers to the considered sample. Refer to the third block in Figure 5 for a graphical representation of the computation. By applying Equation (Equation 2) to all *n* samples, we finally obtain matrix $U_{m}$ of size n×c=60,000×3. The probabilities inside $U_{m}$ do not sum to 1.

Generally, the final predicted class of sample *n* corresponds to the maximum value of the prediction vector returned by the model. For the uncertainty-aware models, this means the maximum among the $u_{n,c,m}$ values in $U_{m}$. Hence, by setting a specific acceptance threshold *t* in range [0,100%], we can force the model to trust only those predictions for which the maximum value is equal to or higher than *t*. If this condition is not satisfied or if more than one class has the same $u_{n,c,m}$ value, the model does not output a predicted class and returns a “not classified” placeholder instead. This behavior is more robust towards uncertain cases for which the samples could belong to more than one class according to their feature predictors.

#### 2.5.3. Bayesian Test

The predictions in $U_{m}$ were arranged in a confusion matrix (one per uncertainty-aware model). This was performed for each considered threshold *t* in range [0,100%] with steps of 10%. By increasing the acceptance threshold, the prediction will be less uncertain, but more samples will be discarded among the total 60,000 samples. To simplify the understanding of results, we applied a Bayesian test following the reasoning in [30] to the data in the confusion matrices. The idea is that asymptomatic samples should be treated as positive for the disease; hence, they should count towards the positive occurrences of diseased samples. The negative test “H” represents the absence of FD according to the molecular test, while the positive test “D” represents the presence of FD according to the molecular test. The probability of obtaining a negative test given that the sample being analyzed by our procedure is of true class “healthy” is P(H|h), while the probability of obtaining a positive test for the same sample is P(D|h). Following the same reasoning, we can obtain the corresponding values for “asymptomatic” (a) and “diseased” (d) samples, which are P(H|a), P(D|a), P(H|d), and P(D|d). By considering P(D) as the incidence of FD on Pinot Noir in the Lombardy region of Italy (estimated equal to 45% of the total number of vineyards according to the literature [54], including both asymptomatic and diseased leaves in a single class), its complementary value P(H) represents the incidence of healthy leaves, equal to 55%. Therefore, we can write:(3)$$\begin{array}{c} P\left(H|h\right)=P\left(h|H\right)\cdot \frac{P(H)}{P(h)}\;P\left(H|a\right)=P\left(a|H\right)\cdot \frac{P(H)}{P(a)}\;P\left(H|d\right)=P\left(d|H\right)\cdot \frac{P(H)}{P(d)}\\ P\left(D|h\right)=P\left(h|D\right)\cdot \frac{P(D)}{P(h)}\;P\left(D|a\right)=P\left(a|D\right)\cdot \frac{P(D)}{P(a)}\;P\left(D|d\right)=P\left(d|D\right)\cdot \frac{P(D)}{P(d)} \end{array}$$

P(h|H), P(a|H), P(d|H), P(h|D), P(a|D) and P(d|D) represent the probability of obtaining a classification output of class “healthy”, “asymptomatic”, or “diseased” given that the sample analyzed is negative (H) or positive (D) to FD. Their value is inferred from the model’s confusion matrix as follows:(4)$$\begin{array}{c} P\left(h|H\right)=\frac{\hat{h_{H}}}{H_{total}}\;P\left(a|H\right)=\frac{\hat{a_{H}}}{H_{total}}\;P\left(d|H\right)=\frac{{\hat{d}}_{H}}{H_{total}}\\ P\left(h|D\right)=\frac{{\hat{h}}_{D}}{D_{total}}\;P\left(a|D\right)=\frac{{\hat{a}}_{D}}{D_{total}}\;P\left(d|D\right)=\frac{{\hat{d}}_{D}}{D_{total}} \end{array}$$
where $\hat{h}$, $\hat{a}$, and $\hat{d}$ represent the occurrences of predicted class “healthy”, “asymptomatic”, and “diseased”, respectively, according to the truth class (“healthy” for subscript “H” and either “asymptomatic” or “diseased” for subscript “D”). $H_{total}$ and $D_{total}$ represent the total number of occurrences that are negative or positive to FD, respectively. Expanding the discussion according to the law of total probability, we can write:(5)$$\begin{array}{cc} P(h) & =P(h|H)\cdot P(H)+P(h|D)\cdot P(D)\\ P(a) & =P(a|H)\cdot P(H)+P(a|D)\cdot P(D)\\ P(d) & =P(d|H)\cdot P(H)+P(d|D)\cdot P(D) \end{array}$$

By using Equations (Equation 4) and (Equation 5) we can calculate the values in Equation (Equation 3). The robustness of the model is expressed by P(H|h), P(D|a), and P(D|d) for the true classes, while misclassifications are represented by the complementary values P(H|a), P(H|d), and P(D|h).

## 3. Results

Referring to the first step of the uncertainty evaluation pipeline in Section 2.5.1, Figure 6 illustrates the values of $\sigma_{1,f,c}$ (light blue) and $\sigma_{2,f,c}$ (pink) for each class. Please note that all values are computed considering the standardized VI values from which predictors are computed (see Section 2.3.2 for details). Hence, the values of $\sigma_{1,f,c}$ and $\sigma_{2,f,c}$ are unitless. The values in Figure 6 are shown per-predictor *f*.

For the “healthy” class, values of $\sigma_{2,f,h}$ are higher than the corresponding $\sigma_{1,f,h}$, while for the “asymptomatic” class, the values are generally equal. For the “diseased” class, most values are similar except for a few predictors for which $\sigma_{1,f,d}$ is higher than $\sigma_{2,f,d}$. The highest variations are observed for the predictors of mARI and mARI2 (anthocyanins) for all three classes. This is due to the formulation of these two VIs (see Section 2.3.2) that include reflectance values measured in the green and red regions of the spectrum. If the anthocyanin content is highly variable inside the leaf, it indicates the presence of both green and red spots; thus, the variability is high in both spectral bands, suggesting a possible contamination from FD. The $\sigma_{2,f,a}$ values of mARI2 predictors are almost doubled for the “asymptomatic” class compared to the corresponding $\sigma_{1,f,a}$, indicating that the synthetic leaves of $D_{\mathrm{synth}}$ generated as explained in Section 2.5.1 may interpolate wrongly the synthetic spectra corresponding to the spectral bands used in the formula. In contrast, mARI is more stable since fewer bands are used. In addition, for the “healthy” class, very low variations are observed for the predictors of RGI and mACI. This is because they both have the green spectral band in the range of 540–580 nm at the denominator in their formulations, which is a spectral band with high normalized reflectance values. For the “asymptomatic” class, values are generally lower compared to the “healthy” class. The highest variations are observed for the “diseased” class, since red areas are more frequent and may be of different sizes. From this analysis alone, it is possible to understand how the three classes are described by the predictors, indicating some differences that may help distinguish asymptomatic samples from the healthy ones.

All models in Section 2.4 were trained on $D_{\mathrm{train}}$ and tested on $D_{\mathrm{Gaussian}}$. Table 4 shows the performance metrics obtained considering only the models’ confidence scores $P_{m}$ (labeled “original”) versus the same metrics obtained considering the uncertainty-aware version of the models (e.g., using $U_{m}$), considering the best acceptance threshold *t*. The performance metrics considered are precision, recall, and F1-score, which are standard metrics used to evaluate ML models’ performance.

For each model, Figure 7 shows (1) the confusion matrix of the “original” model, (2) the confusion matrix of the uncertainty-aware version considering the best acceptance threshold *t*, and (3) a graph showing, for each acceptance threshold set, the prediction accuracy of the uncertainty-aware models divided by class (green, blue, and red solid lines for healthy, asymptomatic, and diseased classes, respectively), and the number of unclassified samples (black line). The reference confidence score resulting from the “original” models (e.g., $P_{m}$) is shown as a colored dashed line for each class. Please note that the prediction accuracies are obtained as the number of true positives over the number of actual samples used (e.g., 60,000 minus the discarded samples, according to the threshold). Referring to the models’ performance metrics shown in Table 4, the impact of uncertainty is evident for the “asymptomatic” class, especially for ML1 and ML4 (uncertainty-aware), which show notably improved metrics. In general, this improvement comes at the expense of the “diseased” class, which consistently shows reduced performance compared to the “original” version of the models. This is also evident from the confusion matrices in Figure 7, highlighting that the composition of the ones obtained for the uncertainty-aware version is very different compared to the “original” models version, even in the case of no acceptance threshold applied (t=0%, meaning that all samples were considered), once again stressing that uncertainty greatly impacts the outcomes of the models. One notable example is in the case of ML1, for which the “original” model classified as “asymptomatic” only 13.2% of true “asymptomatic” occurrences, while the uncertainty-aware version correctly classifies 44.9% of occurrences instead. Moreover, since asymptomatic samples are affected by FD, if the model classifies them as “diseased” it is an acceptable confusion, but classifying them as “healthy” is not. At the same time, confusion between “healthy” and “diseased” samples is not acceptable. With these considerations in mind, the best models are ML1 (Coarse Decision Tree) and ML4 (Ensemble of 30 Bagged Decision Trees), showing a notable increase in the per-class prediction accuracy for the “asymptomatic” class (ML1: 13.2% PA for the “original” model, 44.9% PA for the uncertainty-aware model; ML4: 49.3% PA for the “original” model, 75.7% PA for the uncertainty-aware model). In particular, for ML1, most misclassifications occur between “asymptomatic” and “diseased” samples and vice versa, which is acceptable even if not desirable. A high number of misclassifications also occur for the true class “healthy” confused with “diseased” and “asymptomatic”. This is a conservative case for which the system outputs a high number of false positives. For ML4 (best model obtained for an acceptance threshold set at t=10%), misclassifications between “healthy” and “diseased” classes are still high, but overall, there is a notable improvement in the misclassifications for all the other cases. It is worth noting that the unclassified samples for ML3 and ML4 are 26% and 21%, respectively. The per-class reductions are 4% (ML3) and 1% (ML4) for the “healthy” class, 11% (ML3) and 7% (ML4) for the “asymptomatic” class, and 62% (ML3) and 56% (ML4) for the “diseased” class.

The results of the Bayesian test obtained by following the procedure in Section 2.5.3 are shown in Table 5 for each uncertainty-aware model. The Table also shows with colored arrows how to interpret the values (e.g., green arrow up: better performance when values are close to 100%; green arrow down: better performance when values are close to 0%; similar reasoning applies for red arrows). Considering the best-performing models ML1 and ML4, the superior performance of ML4 is evident except for the misclassifications between “diseased” and “healthy” samples, as highlighted by the low value of P(D|d) and high value of P(H|d). In addition, the performance of ML3 (which has the best values for four out of six Bayesian probabilities) can be explained by considering the number of unclassified samples (reduced by 26% against the 21% reduction in the case of ML4). However, by considering the misclassifications highlighted by the confusion matrix, it is evident that this model is suboptimal since almost half of the samples of classes “healthy” and “asymptomatic” are confused with the “diseased” class.

### 3.1. Discussion

By taking into account the findings of our previous works [30,31] to complement the variations shown in Figure 6, we can observe that the most useful VIs to distinguish between “healthy” samples and the other two are mARI, mARI2, ACI, and mACI. The drawback is that, for these four VIs, the differences between “asymptomatic” and “diseased” samples are minimal. In contrast, RGI and CI VIs maximize differences between “diseased” and the other two, but differences among “healthy” and “asymptomatic” samples in this case are minimal. A biological reason for this behavior can be found in the VIs definition; in fact, mARI, mARI2, ACI, and mACI are all indicators representing anthocyanin content, responsible for the red pigments in the leaves (red spots are a symptom of FD in the case of red grape varieties). Red pigments are more prominent in the case of “diseased” samples, while there may be just a few in the case of “asymptomatic” samples. RGI and CI VIs are representative of how many green areas are present in the leaf, so evidently their values are higher in the case of “healthy” samples. Since “asymptomatic” samples are mostly green as well, these two indicators appear similar for both classes.

Other interesting conclusions can be drawn by looking at the graphs in the third column of Figure 7. For example, in the case of ML5 (Wide Neural Network) it is evident by the trend of the unclassified samples (black line) that only setting the threshold to 10% and 90% produces a variation in the models, meaning that for most *n* samples the corresponding *u* values (e.g., the probabilities contained in the corresponding $u_{n}$ vectors) are in the range [0.1, 0.9]. For ML2 (Naive Bayes), increasing *t* reduces the number of samples from all the classes at the same time, keeping the overall prediction accuracy of the three classes almost the same. In the case of ML4, by looking at the per-class trends (green, blue, and red lines), we can observe that the most uncertain cases that become “unclassified” samples (black line) are the “diseased” ones, since the red line decreases while the other two increase with the increase in the acceptance threshold *t*. The same conclusions can be drawn for ML1, while for ML3 (k-NN) the “diseased” correct classifications increase while the other two decrease with the increase in *t*. Therefore, incorporating uncertainty into the decision process with a suitable acceptance threshold is a good way to improve the robustness of the models, as only samples with low uncertainty will be classified as for the best-performing model ML4 with t=10%. Interestingly, the performance achieved by original models is often reduced when including uncertainty, because the phenomenon is complex. The rationale is that, instead of producing incorrect outputs with high confidence, or even correct predictions that are not explainable, it is best to give room for doubt and eventually prompt the researcher to test the sample with more accurate tests (e.g., PCR).

Now considering the Bayesian test results shown in Table 5, it is observed that the false positives are an issue for all models, as highlighted by the values of P(H|d) that are higher than 50% in all cases. Similarly, all models show low performance in the correct detection of true positives for the “diseased” class, P(D|d), mostly because samples of this class get confused as “asymptomatic” and sometimes as “healthy” (like in the case of ML5). On the other hand, several models show good performance for both P(H|h) and P(D|a).

The results in Figure 7 and Table 4 and Table 5 highlight that Decision Trees and their variants are probably the best models to tackle the problem under study; however, there is room for improvement to reduce the number of false positives. For this model architecture, our approach demonstrated that incorporating uncertainty in the decision process produces a model more robust toward uncertain cases, and, in turn, the detection rate of asymptomatic samples is notably improved (from 49.3% PA achieved by the “original” model to 75.5% PA achieved by its uncertainty-aware version with t=10%).

### 3.2. Limitations of the Study

The main drawback of the presented study is that the acquired dataset is limited in size to only 201 hypercubes. This issue was partially addressed through the Monte Carlo generation explained in Section 2.5; however, an extensive validation of our conclusions should be conducted on new data collected in the field. The main reason for the low number of samples collected is that the commercial vineyard participating in this research was small, and not all plants were affected by FD. Furthermore, since FD is a quarantine disease, vine-growers must apply treatments, thus reducing potential infection risks. Another reason is that it is impossible to know in advance which plants will be affected by FD, as contagion occurs during winter, but symptoms appear in late summer. For all these reasons, the high number of leaf samples analyzed was mostly useless, limiting the actual size of the dataset (also considering that we wanted to balance the classes by having the same number of samples each). Another limitation is that only one variety was tested; therefore, our conclusions may not be applicable to other red grape varieties. In addition, since symptoms vary for white grape varieties, a different set of VIs must be chosen to properly characterize the disease (e.g., carotenoids should be analyzed instead of anthocyanins, which are responsible for the yellow pigments). In the future, we aim to extend the size of the collected dataset and also conduct specific experiments on white grape varieties.

Another point of discussion is represented by the machine learning analysis. First, the selected feature predictors were chosen based on the results of our previous works [30,31]. In the case of red grape varieties, FD symptoms affecting the leaves mainly include red spots of various shapes, sizes, and positions on the leaf’s surface. By using quartiles as statistical indicators, we can roughly represent how the population of VIs values is distributed in the leaf. Ideally, samples that are healthy or heavily affected by FD exhibit uniform coloration (either green or red); thus, the distributions of their VIs have low variance and small quartile ranges. In contrast, asymptomatic samples are characterized by greater VIs’ variance due to the presence of a few discolored portions or areas with a lower intensity response in the green spectral band (which could be linked to a reduced chlorophyll content due to FD symptoms impeding photosynthesis through starch accumulation [55]). Distribution’s skewness and kurtosis were not considered since they were less effective compared to quartile ranges, as demonstrated in our previous publications. Second, since the focus of this work was to demonstrate how uncertainty-aware models can aid FD research by properly addressing uncertain cases (represented by asymptomatic samples), we limited our analysis to a small subset of ML models. They were chosen as representatives of their categories, also considering whether their output was suitable for our aggregation method as described in Section 2.5.2 (for which confidence scores are necessary). A more in-depth analysis of our approach’s performance when applied to other models, such as CNNs, will be a topic for the future.

Finally, since the hyperspectral camera used for this work is suited for laboratory experiments, acquiring a hypercube is a task that requires a few seconds (mostly due to the density of the spectral signal measured). Therefore, it is not suitable for on-the-go applications based on land rovers or UAVs that move around the field. However, this study gives insights to the research community so that specific spectral bands can be selected instead of the whole signal. This information is crucial for the choice of a suitable multispectral camera, which acquires data faster due to a reduced density of the measured signal. Moreover, the SWIR region was not measured in our experiments; thus, we are lacking information about other biological indicators that can be observed in that region. The main reason for this choice was that the literature background [9,10,11,12,13] stressed how most information related to FD detection can be observed in the VIS-NIR range instead.

## 4. Conclusions

This paper describes a novel method for the classification of leaf samples affected by the phytoplasma of Flavescence dorée (FD) into three classes (“healthy”, “asymptomatic”, and “diseased”). Compared to the literature, the challenge of early identification of asymptomatic samples was never approached before with a contactless method, nor was the in-depth uncertainty of classification outcomes. A total of 201 samples, equally divided between the classes, were collected from a field of Pinot Noir located in northern Italy in 2023 and 2024. The measurement is based on the acquisition of hypercubes through a hyperspectral camera. Since the dataset size is not sufficient to properly estimate uncertainty on the input data, we addressed this issue in two steps by exploiting a double Monte Carlo procedure in compliance with the “Guide to the Expression of Uncertainty”. We propose the application of uncertainty-aware models that aggregate the original confidence scores of the models with statistical indicators returned by a z-test. These models can understand if a sample fits multiple categories at the same time and can eventually produce an “unclassified” outcome instead of guessing the class when the uncertainty is too high. Our structure can be applied on top of a plethora of ML and DL models avoiding the need of modifying the models’ original structure. The results highlight how our method notably improves recognition of “asymptomatic” samples with respect to the original models without uncertainty applied. The best model structure is the Decision Tree, achieving a prediction accuracy for “asymptomatic” samples of 75.7% against the original 49.3%. However, this results in a slight decrease in the prediction accuracy for the other two classes and an overall reduction in the samples (by setting an acceptance threshold of 10%, the “unclassified” samples were 1%, 7%, and 56% for “healthy”, “asymptomatic”, and “diseased” classes, respectively). Our future directions include (1) the adoption of image processing techniques to understand if there are distinctive spatial patterns that can be used as predictors, (2) extensive tests on more ML and DL models such as CNNs, and (3) the collection of new data to increase the dataset size and test more grape varieties.

## Figures and Tables

**Figure 1 sensors-25-07493-f001:**
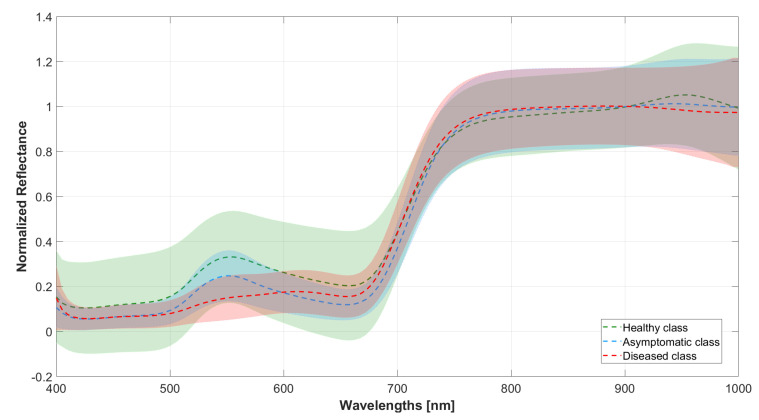
Plot of the average normalized spectra of the processed leaf hypercubes shown per class. The solid line refers to the mean value, computed as the mean over the 67 samples of the class, while the shaded area refers to the per-band standard deviation.

**Figure 2 sensors-25-07493-f002:**
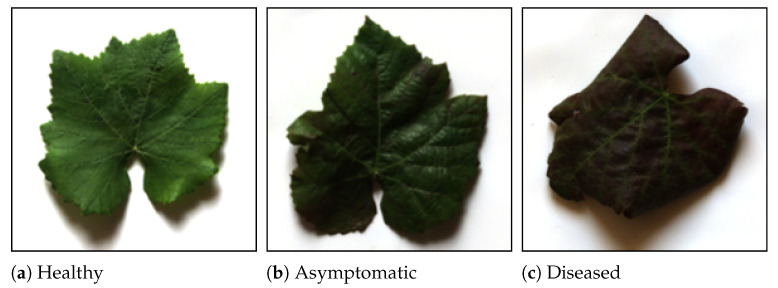
Image examples of the collected leaves. (**a**) Healthy sample, collected in July; (**b**) asymptomatic sample, collected in July from a plant that tested positive for FD in August; (**c**) diseased sample, collected in August from the sample plant of sample (**b**).

**Figure 3 sensors-25-07493-f003:**
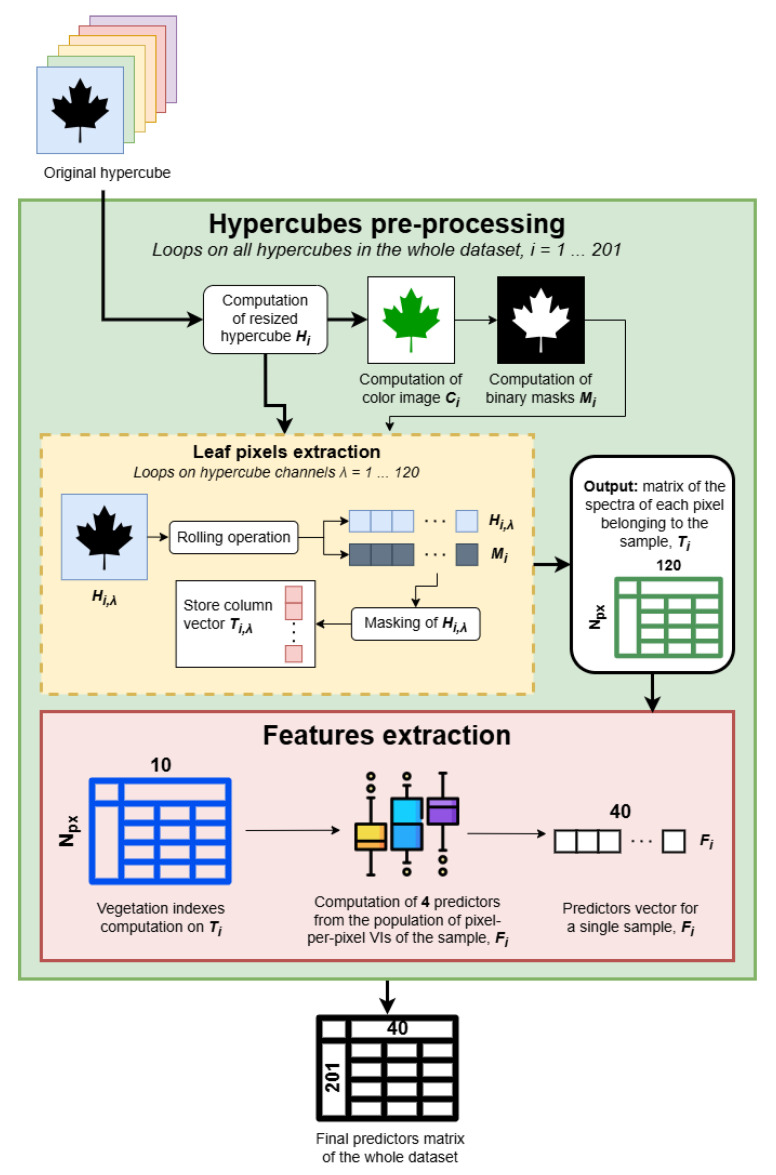
Scheme of the pre-processing procedure in detail. Big solid arrows represent multidimensional data, while thin arrows represent simple data like images or vectors.

**Figure 4 sensors-25-07493-f004:**
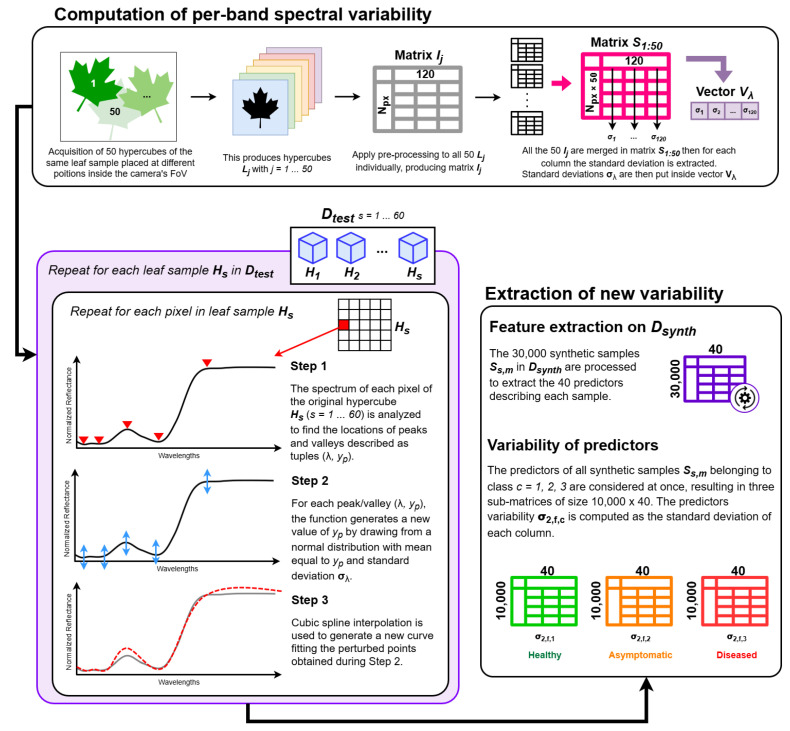
Representation of the generation of data in $D_{\mathrm{synth}}$, used to compute $\sigma_{2,f,c}$. Please note that the generation of synthetic leaves is repeated M=500 times for each original sample in $D_{\mathrm{test}}$ and for each pixel.

**Figure 5 sensors-25-07493-f005:**
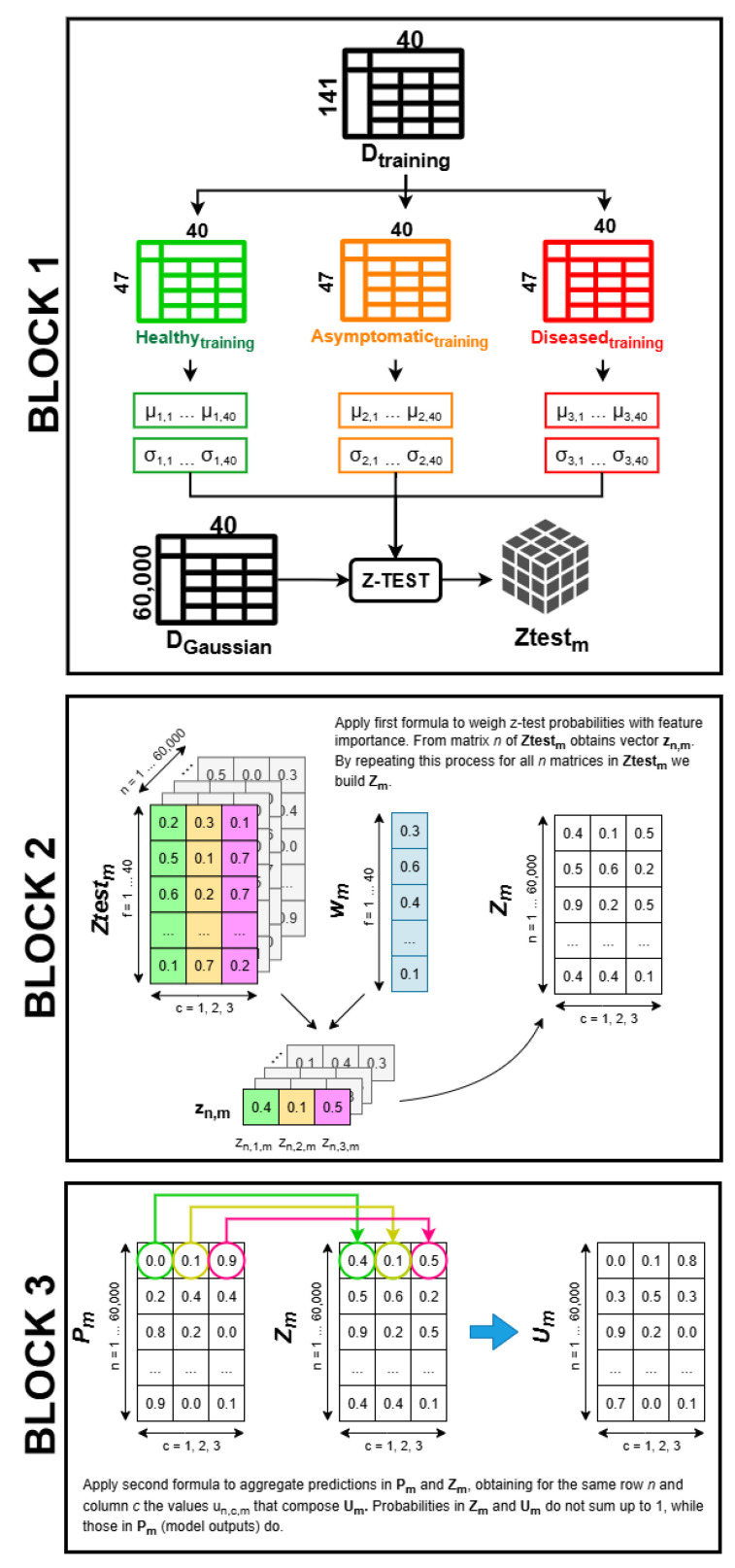
Scheme of the procedure conducted to obtain predictions $U_{m}$ of each uncertainty-aware model.

**Figure 6 sensors-25-07493-f006:**
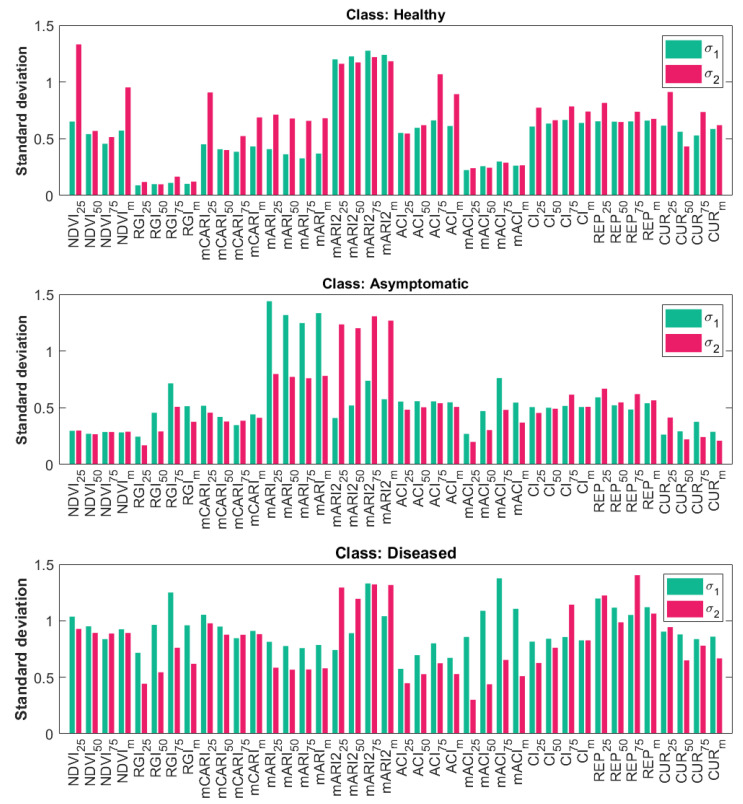
Bar plots showing the values of $\sigma_{1}$ (light blue) and $\sigma_{2}$ (pink) for all predictors and classes.

**Figure 7 sensors-25-07493-f007:**
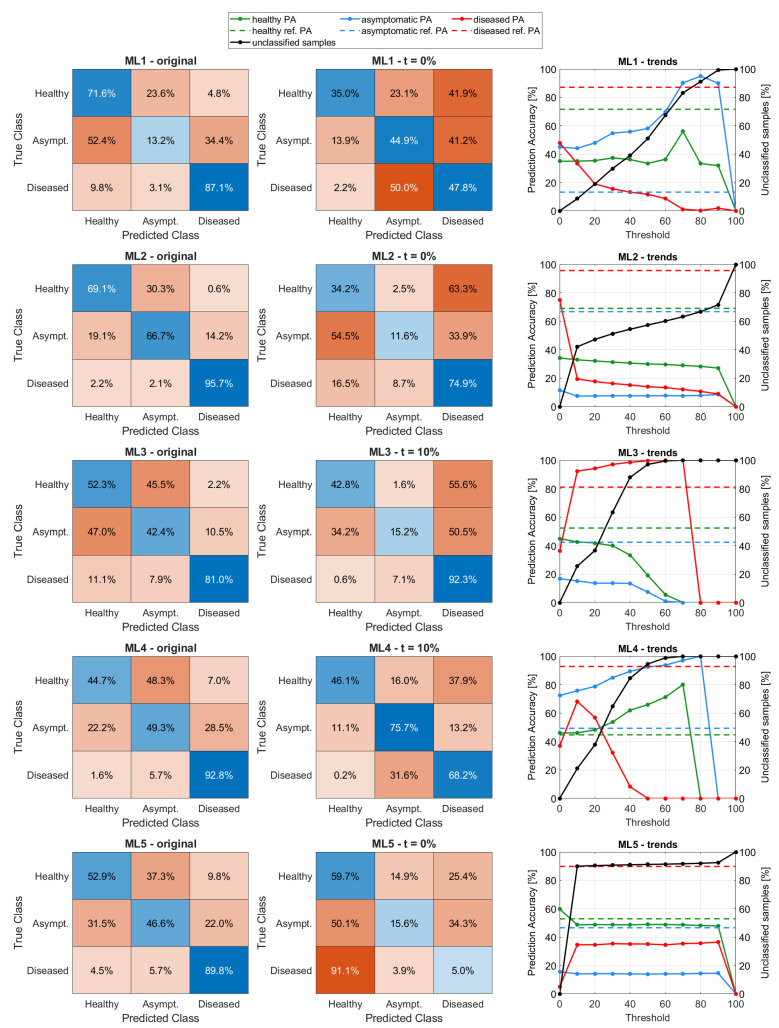
Plots showing, for each threshold set, (1) the prediction accuracy (PA) of the uncertainty-aware models per class (green, blue, and red solid lines representing healthy, asymptomatic, and diseased classes, respectively) and (2) the per-class corresponding reference PA computed from the “original” model (green, blue, and red dashed lines representing healthy, asymptomatic, and diseased classes, respectively), (3) the number of unclassified occurrences (black line).

**Table 1 sensors-25-07493-t001:** Summary of the composition of the datasets, showing the number of samples per class according to the acquisition campaign.

Dataset	Healthy	Asymptomatic	Diseased	Total
Campaign 1	10	45	45	100
Campaign 2	57	22	22	101
Total	67	67	67	201
$D_{\mathrm{train}}$	47	47	47	141
$D_{\mathrm{test}}$	20	20	20	60

**Table 2 sensors-25-07493-t002:** Summary of the vegetation indexes adopted in this work. λ refers to the spectral reflectance measured at a specific wavelength, while the overline indicates that the spectral range indicated was averaged to take into account the contribution of all the spectra considered. All VIs are unitless by definition.

Formulation	Description
$NDVI=\frac{\bar{\lambda 760:800}-\bar{\lambda 658:678}}{\bar{\lambda 760:800}+\bar{\lambda 658:678}}$	This is a common index used to determine the presence and health of vegetation. It ranges from −1 (water or clouds) to +1 (dense vegetation). Values closer to 0 represent urban areas or bare soil [19].
$RGI=\frac{\bar{\lambda 658:678}}{\bar{\lambda 540:580}}$	It is used to assess vegetation health and the density of vegetation areas. High values indicate healthier or denser vegetation, while lower values suggest non-vegetated areas or sparse vegetation [19,37].
$CI=\frac{\bar{\lambda 760:800}}{\bar{\lambda 707:727}}-1$	This index is used to estimate the chlorophyll content in leaves, which is responsible for green pigmentation and plays a crucial role in photosynthesis. High CI values indicate high chlorophyll content [19,38].
$mCARI=1.5\cdot \frac{2.5\cdot (\lambda 800-\lambda 670)-1.3\cdot (\lambda 800-\lambda 550)}{\sqrt{{\left(2\cdot \lambda 800+1\right)}^{2}-\left(6\cdot \lambda 800-5\cdot \lambda 670\right)-0.5}}$	This index estimates chlorophyll content in leaves when other indexes fail due to disturbance factors such as soil background or canopy structure. It is formulated to minimize the influence of these disturbance factors by incorporating multiple wavelengths [39].
$REP=700+40\cdot \frac{\mathrm{pRE}-\lambda 700}{\lambda 740-\lambda 700}$	This VI is used to estimate the position of the red-edge band, where rapid changes in reflectance correspond to changes in vegetation health, chlorophyll content, and physiological status [39]. The formulation includes a parameter called “pseudo-Red-Edge” (pRE), which represents the mid-point between two wavelengths of the red-edge spectral band: pRE=(λ670+λ780)/2.
$CUR=\frac{\lambda 675\cdot \lambda 690}{\lambda 683^{2}}$	Since the red-edge region is sensitive to changes in chlorophyll content and vegetation structure, measuring the curvature of the spectral reflectance curve in the red-edge region can be a tool to analyze plant physiological processes and health. Higher values correspond to big changes in chlorophyll content or vegetation structure [39].
$mARI=\left(\frac{1}{\lambda 550}-\frac{1}{\lambda 700}\right)\cdot \bar{\lambda 760:800}$	This VI determines the presence and concentration of anthocyanins (responsible for red, purple, and blue pigments) in plant leaves. Higher values indicate higher anthocyanin content in leaves [19,37].
$mARI2=\frac{1}{\bar{\lambda 540:580}}-\frac{1}{\bar{\lambda 707:727}}\cdot \bar{\lambda 760:800}$	This is a modified version of mARI that takes into account multiple wavelengths of the green and red-edge spectral bands [30]. Ideally, this index should provide an averaged and more accurate estimation of the concentration of anthocyanins in the leaf.
$ACI=\frac{\bar{\lambda 540:580}}{\bar{\lambda 760:800}}$	This VI measures the quantity of anthocyanin in leaves and is sensitive to its changes. Higher values generally indicate higher concentrations of anthocyanins in plant leaves [19,37].
$mACI=\frac{\bar{\lambda 760:800}}{\bar{\lambda 540:580}}$	This VI is designed to provide a more robust estimation of anthocyanin content than ACI since the NIR band is sensitive to factors other than anthocyanin content (e.g., leaf structure, water content) [19,40].

**Table 3 sensors-25-07493-t003:** Relevant parameters used for the synthetic variability computation (Section 2.5.1) and classification uncertainty estimation (Section 2.5.2).

Param.	Description	Value
*c*	Class index corresponding to “healthy”, “asymptomatic”, and “diseased”.	3
*f*	Feature predictors computed as described in Section 2.3.2.	40
λ	Spectral band index, corresponding to 400–1000 nm (step of 5 nm).	120
*s*	Number of samples contained in $D_{\mathrm{test}}$, corresponding to its rows.	60
*i*	Number of samples per class contained in $D_{\mathrm{test}}$.	20
$N_{px}$	Number of valid pixels belonging to the leaf depicted in a certain hypercube. This number depends on the size of the leaf.	100,000∼300,000
*M*	Number of Monte Carlo simulations conducted to generate synthetic hypercubes as described in Section 2.5.1.	500
*j*	Number of hypercubes collected to compute the spectral variability as described in Section 2.5.1.	50
*m*	Number of tested models according to Section 2.4.	5
*G*	Number of Monte Carlo simulations conducted to generate synthetic predictors as described in Section 2.5.2.	1000
*n*	Number of rows in $D_{\mathrm{gaussian}}$, obtained by generating Monte Carlo predictors *G* times for each original sample *s* in $D_{\mathrm{test}}$ as described in Section 2.5.2.	60,000

**Table 4 sensors-25-07493-t004:** Performance metrics for each model, divided per class. Values in bold are the highest for the specific metric among the two versions of the model (original versus uncertainty-aware considering the best acceptance threshold).

Models	Healthy			Asymptomatic			Diseased		
	**Precision**	**Recall**	**F1-Score**	**Precision**	**Recall**	**F1-Score**	**Precision**	**Recall**	**F1-Score**
ML1 Orig.	53.5%	71.6%	61.2%	33.1%	13.2%	18.9%	69.0%	87.1%	77.0%
ML1 t = 0%	68.5%	35.0%	46.3%	38.1%	44.9%	41.2%	36.5%	47.8%	41.4%
ML2 Orig.	76.4%	69.1%	72.6%	67.3%	66.7%	67.0%	86.6%	95.7%	90.9%
ML2 t = 0%	32.5%	34.2%	33.3%	50.9%	11.6%	18.9%	43.5%	74.8%	55.0%
ML3 Orig.	47.4%	52.3%	49.7%	44.3%	42.4%	43.3%	86.4%	81.0%	83.6%
ML3 t = 10%	55.2%	42.8%	48.2%	63.6%	15.2%	24.6%	46.5%	92.3%	61.9%
ML4 Orig.	65.3%	44.7%	53.1%	47.7%	49.3%	48.5%	72.3%	92.7%	81.3%
L4 t = 10%	80.3%	46.1%	58.6%	61.4%	75.7%	67.8%	57.2%	68.2%	62.2%
ML5 Orig.	59.5%	52.9%	56.0%	52.0%	46.6%	49.1%	73.8%	89.8%	81.0%
ML5 t = 0%	29.7%	59.7%	39.7%	45.3%	15.6%	23.2%	7.7%	5.0%	6.1%

**Table 5 sensors-25-07493-t005:** Results of the Bayesian test for each uncertainty-aware model, divided per class (h: healthy class, a: asymptomatic class, d: diseased class). Values in bold are the highest per row. Green arrow up: better performance when values are close to 100%; green arrow down: better performance when values are close to 0%; red arrow up: worse performance when values are close to 100%; red arrow down: worse performance when values are close to 0%.

Probability	ML1	ML2	ML3	ML4	ML5
P(H|h) ↑ ↓	84%	54%	68%	88%	51%
P(D|a) ↑ ↓	63%	77%	87%	76%	35%
P(D|d) ↑ ↓	47%	41%	48%	40%	39%
P(H|a) ↑ ↓	37%	23%	13%	24%	65%
P(H|d) ↑ ↓	53%	59%	52%	60%	61%
P(D|h) ↑ ↓	16%	46%	32%	12%	49%

## Data Availability

The raw data supporting the conclusions of this article will be made available by the authors on request.

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
