# Peer review of "Incorporating Uncertainty in Machine Learning Models to Improve Early Detection of Flavescence Dorée: A Demonstration of Applicabilityâ€"

_sensors, 2025, doi:10.3390/s25247493_

Round 1
Reviewer 1 Report
Comments and Suggestions for Authors
General comments:
The study uses HSI and uncertainty-aware methods to address a significant problem in the early detection of Flavescence dorée. Although novel, the extensive use of artificial datasets (60,000 features and 30,000 spectra) raises questions regarding overfitting and biological representativeness. It's unclear how natural spectrum variability was maintained due to the Monte Carlo procedure's lack of information. Confidence in robustness is limited by validation that is mostly based on synthetic data and Bayesian testing. To support assertions of decision tree superiority, more extensive algorithmic comparisons and outside validation are required. Thus, this study needs massive improvements or claerificatons.
Specific comments:
L#02: Kindly reconsider this sentence.
L#04: “Combination of the HSI and VIs”, this sentence is absurd because the VIs are calculated using the HSI data, it is not an independent entity.
Add numerical results in the abstract section.
The basic dataset is very less, also for the creation of the synthetic dataset.
Has the study conducted any physicochemical analysis to validate the presence of the “Flavescence dorée”
Please properly name the ML algorithms, no need to code them as ML1, ML2…
Separate the results and discussion sections.
Add more ML analysis regarding important features relevant to Flavescence dorée.
Show the real reflectance for each class.
The data is not fully explored.
Properly restructure the manuscript.
Figure 5. Move it to the results section.
Shorten the conclusion section.
Author Response
Q1: The study uses HSI and uncertainty-aware methods to address a significant problem in the early detection of Flavescence dorée. Although novel, the extensive use of artificial datasets (60,000 features and 30,000 spectra) raises questions regarding overfitting and biological representativeness. It's unclear how natural spectrum variability was maintained due to the Monte Carlo procedure's lack of information. Confidence in robustness is limited by validation that is mostly based on synthetic data and Bayesian testing. To support assertions of decision tree superiority, more extensive algorithmic comparisons and outside validation are required. Thus, this study needs massive improvements or claerificatons.
A1: The aim of the study was to provide a novel point of view in FD research by taking into account the metrological uncertainty associated with the measurement (from data acquisition up to class prediction), not to present our method as the conclusive solution to solve the issue of FD detection in general. This is why we presented our method as a demonstrative application even if the number of acquired samples is relatively low (201 hypercubes). Evidently, the presented method must be validated with new in-field data; this work serves only as the stepping stone toward a comprehensive approach to at least complement FD detection in early stages of growth/asymptomatic samples before conducting lab tests on the leaves DNA. We changed the article’s title to better reflect this aspect.
We failed to understand what the reviewer’s meant by “Monte Carlo procedure lack of information”, since the statistical method is well known in literature and we provided all the necessary information to apply it both to generate Dsynth (Section 2.5.1) and Dgaussian (Section 2.5.2) and even supplied two figures to illustrate the process in full. The procedures were done in compliance with standard measurement science techniques, especially the “Guide to the expression of Uncertainty in Measurement” standard procedure no. 101:2008 published by the Joint Committee for Guides in Metrology, as stated at L#371 (ref. 49). To improve understanding of the process rationale and also provide a short summary of the steps, we improved the initial portion of Section 2.5 and slightly modified Fig. 2. To summarize its contents, we now make clear that the procedure tries to address two issues: (1) the low number of data in the dataset, insufficient for a proper modelization of the input data uncertainty, and (2) the issue of uncertainty propagation inside the model, which closely depend on the model’s structure. To address issue (1), we came up with a double Monte Carlo procedure that first generates synthetic hypercubes (see Section 2.5.1) from which we compute the variability of the resulting feature predictors. This variability is referred to σ2 (per feature and per class). σ2 is compared with σ1, computed on the original data in Dtest. If the two are comparable, this means that the procedure was successful and thus σ2 can be considered a reliable proxy of the input data uncertainty. Refer to Fig. 4 for the results of this comparison. Then, we generate again new samples, this time using σ2 as the input variability to generate predictors directly from Dtest (see Section 2.5.2). This is a faster way of generating a synthetic dataset (compared to the hypercubes generation) while keeping it statistically reliable compared to the original dataset. The generated samples in the new Dgaussian are then used as test data. To address issue (2), we adopted the aggregation system described in refs 47-48. In contrast with the original approach that propagates uncertainty inside the model, we aggregate the confidence scores produced as output by the model with the outcome of a ztest statistical operation. The two are aggregated using the formulation in Eq. 1 and 2, as described in Section 2.5.2.
Biological representativeness of the generated samples is retained in both cases, since to generate Dsynth data we simply modified the spectra of each pixel of the leaf of a quantity corresponding to the specific standard deviation at each wavelength as measured by the 50-hypercubes repetitions test (Section 2.5.1). In the case of Dgaussian, data generation of the feature predictors was conducted again considering as standard deviation the variability of each predictor per class resulting from the previous analysis conducted on Dsynth (σ2). Comparison shown in Fig. 4 highlights that the method is robust since the two variabilities (σ1 of original predictors in Dtest versus σ2 of Dsynth predictors obtained after generating the hypercubes) are comparable for all the feature predictors and for all three classes. We agree that some generated data points could be unrealistic from a biological point of view, however we believe that a strong statistical approach as the one we applied will limit these occurrences to just a few and thus not alter the results dramatically. Therefore, by using σ2 to generate the 40 predictors in Dgaussian, the eventual inaccuracies due to the spectra generation are reduced.
As for the classification results and Bayesian test, numerical results evidently depend on the quantity of the original dataset (201 hypercubes), which is the drawback of the presented method that was never kept hidden (instead, it was clearly stated several times in Section 2.5). However, the method itself is statistically solid (again, refer to ref. 49) and if applied to a new bigger dataset the results will prove significant for FD research. This is the reason why the article focus is on the uncertainty estimation methodology instead of the specific conclusions tied to plant biology (which could be drawn with confidence if more data are used, thus strengthening the numerical stability of the results), also stressed by the new article title. As such, we limited the models under test since the aim was not to find the best ML algorithm to obtain maximum classification accuracy. Models of choice were selected according to our past experiences published in ref. 30-31 and on the models’ capacity to produce a “confidence score” as output (e.g., SVMs do not, so our method described in Section 2.5.2 cannot be applied), hence we do not provide extensive comparisons with all existing methods because it was not our aim. The superiority of decision trees is thus demonstrated only in this example and are not true in general. As for validation with in-field data, we agree that this is necessary but it is a process that takes time. These issues are stressed in the “Limitations of the study” section to improve clarity.
Q2: L#02: Kindly reconsider this sentence.
A2: We believe the sentence noted by the reviewer is “…conventional molecular laboratory tests are unable to detect the presence of the phytoplasma in asymptomatic samples”. Current tests conducted in specialized laboratories are mainly based on PCR and require the sample under test to be heavily subjected to symptoms of the disease to detect the phytoplasma. Here, we mainly refer to paid services outside of research centers located in our country. In addition, if the leaf sample is relatively young (e.g. collected at most in early July), the presence of the phytoplasma genome will be low, thus making it difficult for standard practices to detect it. Nonetheless, we removed this sentence to avoid conflict; instead, we better explain our point in the Introduction Section.
Q3: L#04: “Combination of the HSI and VIs”, this sentence is absurd because the VIs are calculated using the HSI data, it is not an independent entity.
A3: We modified this sentence as follows: “…by proposing a methodology exploiting per-pixel data obtained from hyperspectral imaging to produce…”
Q4: Add numerical results in the abstract section.
A4: We modified the abstract as requested.
Q5: The basic dataset is very less, also for the creation of the synthetic dataset.
A5: We understand the reviewer’s concern about the dimension of the acquired dataset and we agree (it was, in fact, already stated in the text and it’s also the reason why we developed a complicated procedure to estimate uncertainty, as described in the modified version of Section 2.5). However, collecting samples is not easy and it’s a slow process requiring collaborations with commercial vineyards and at least a whole season to observe the growth of the plants. We also wish to stress that the collected data was taken from a commercial vineyard, meaning that it was not possible to collect samples from every plant but only from a limited subset due to the vineyard owners’ requests. Moreover, commercial vineyards perform FD mandatory treatments regularly, so it is not certain if the collected samples over the season will result positive to FD and this impacts the quantity of samples in the dataset. Frequently we had to discard samples that apparently seemed to be affected by FD but after PCR analysis resulted negative. Nonetheless, in 2025 we started another campaign in collaboration with a different vineyard and are (slowly but surely) collecting new samples of different varieties there.
With respect to the synthetic dataset, we wish to stress that we first collected 50 hypercubes of the same sample (not presented in any dataset) to study the predictors’ variability observed by our instrumentation (Section 3.1) in compliance with measurement science practices. Synthetic generation is therefore conducted by exploiting this variability to generate new realistic data and their reliability is assessed by comparing the variability of the original data σ1 to that of the synthetic data σ2 (see Fig. 4). Finally, classification uncertainty was obtained following the combined Z-score/Confidence method presented in Section 2.5.2 (see Fig.3). All procedures comply with the “Guide to the expression of uncertainty in measurement” standard procedure no. 101:2008 published by the Joint Committee for Guides in Metrology, as already stated at L#317 (ref. 48). However, it is still worth noting that the size of the dataset is a limitation to our study, hence we address this issue in the “Limitations of the study” section.
Q6: Has the study conducted any physicochemical analysis to validate the presence of the “Flavescence dorée”
A6: As already stated at the end of Section 2.2, after collecting the leaf samples they were sent to an external laboratory to certify the presence or absence of FD using PCR. In our country, this is the standard protocol adopted to verify the presence of the phytoplasma on leaf samples. We did not conduct the laboratory analysis ourselves (it was a paid service).
Q7: Please properly name the ML algorithms, no need to code them as ML1, ML2…
A7: We understand that reading codenames for the tested machine learning models could be confusing. However, we prefer it this way since using only the model name (e.g. wide neural network) it’s limiting; in fact, a model is not only defined by the algorithmic structure adopted but also on its hyperparameters. This is why we typically describe models in a separate section alongside their parameters so that readers can easily be redirected to that section for further details and reproducibility. This is also a standard practice in measurement science and computer vision research. Therefore, we prefer to keep the codenames in the article.
Q8: Separate the results and discussion sections.
A8: We thank the reviewer for this suggestion. We separated the sections as requested.
Q9: Add more ML analysis regarding important features relevant to Flavescence dorée.
A9: The work presented in the submitted manuscript comes after two previously published works (ref. 30-31) in which we explored different ML models; hence, in this article we limited our study to the most promising ones and instead focused on the evaluation of uncertainty related to both the measure and the prediction. Moreover, not every ML model can be used with the proposed uncertainty-aware structure, because it requires a probabilistic “confidence score” (e.g. SVMs do not provide scores but instead they provide support vectors). Therefore, such models were discarded and not tested in this study, where the focus was on the uncertainty estimation instead of the classification metrics. We added this reasoning in Section 2.4 as well to improve clarity. Moreover, features relevant for FD detection may vary according to the variety (e.g. anthocyanin content is relevant for red grape varieties but may not be as relevant for white grape varieties, since we didn’t test them yet) and to the presence or absence of the pathogen. We discuss this issue in the new “Limitations of the study” section.
Q10: Show the real reflectance for each class.
A10: According to the camera’s manufacturer guidelines (HERA VIS-NIR produced by NIREOS), intensity calibration of the camera must be conducted every time the camera is used to properly adjust the measured quantities according to the experimental set-up conditions, because the measurement technology is based on Fourier Transform. This is why in our work we only used normalized reflectance that is obtained after intensity correction. Thus, the “real reflectance” the reviewer is referring to (which we think it is the absolute reflectance measured without normalization) cannot be used since it’s not reliable for this specific instrument.
Q11: The data is not fully explored.
A11: We unfortunately failed to understand the reviewer’s comment. If possible, can you provide more information regarding how our method does not explore properly the acquired data and how this relates to the focus of the article (which was the understanding of measurement uncertainty associated with the whole pipeline, e.g. data acquisition and class prediction jointly)? We wish to stress that this article was not a data science paper for which the main focus is to extract a huge amount of information and analyze it using every existing method, nor we aimed to draw conclusions on plant biology or toxicology. If the reviewer has specific concerns about our method or suggestions about approaches that were not tested to explore the acquired data better, we will gladly try to answer them in the future.
Q12: Properly restructure the manuscript.
A12: We revised the manuscript by including Section 3 under the “Materials and Methods” section and now adding a separate “Limitations of the approach” sub-section after the results. We also wish to stress that the structure adopted complies with the journal’s guidelines.
Q13: Figure 5. Move it to the results section.
A13: Figure 5 is simply too big and LateX moved it where it was possible to place it in a seamless way. After the modifications made following the reviewers suggestions, Figure 5 is now properly placed in Section 3.
Q14: Shorten the conclusion section.
A14: We reduced the word count of the Conclusions sections as requested.
Reviewer 2 Report
Comments and Suggestions for Authors
Early detection of Flavescence dorée leaf symptoms remains an open question for the research community, as conventional molecular laboratory tests are unable to detect the presence of the phytoplasma in asymptomatic samples. This work tries to fill this gap by proposing a methodology that combines hyperspectral imaging and vegetation indices to produce features suitable for machine learning training. The models’ predictions were further examined by a Bayesian test, demonstrating that with this approach, the best models belonged to the class of decision trees. However, there are still some issues with the manuscript that need to be addressed urgently.
- Line 228:Is a total of 5 ML appropriate? Additionally, the solutions proposed below for the 5 ML are all feasible. We recommend the authors consider them!
- Many parameters in Table 2 have overly complex and lengthy subscripts. We recommend the authors simplify them. For example: λ540:580.
Author Response
Q1: Line 228:Is a total of 5 ML appropriate? Additionally, the solutions proposed below for the 5 ML are all feasible. We recommend the authors consider them!
A1: We thank the reviewer for this comment. We wish to stress that the focus of this article is not the test and comparison of every existing ML model structure but, instead, to demonstrate the applicability of our method (which builds upon the existing ML model) and how including uncertainty allows to express the intrinsic complexity of a phenomena such as FD detection. We chose a limited number of ML models among the available and suitable ones in MATLAB, also to keep processing times reasonable. The core of our method (described in Section 2.5.2) is easily reproducible and applicable to other suitable ML architectures, but keep in mind that not every ML model can be used, since our method depends on whether the model outputs a probabilistic “confidence score” or not (e.g. SVMs output support vectors, not probabilities). Confidence scores are necessary to build the uncertainty-aware output described in Section 2.5.2. We stressed once again this matter in Section 2.4 to improve clarity. As for the outcomes of the models presented in Figure 5, it is highlighted how introducing uncertainty positively impact the model’s performance only in the case of ML4 (Decision Tree), while in the other cases overall performance is reduced (see second column of the figure; the first column shows the models’ performance without uncertainty applied). Please note that the goal was to improve detection of asymptomatic samples (blue lines in the graphs in column 3 of Fig. 5). Interestingly, the performance achieved by the original models (without uncertainty) is often reduced when including uncertainty because the phenomena is complex, thus overall performance on the new generated data gives room for doubt (e.g., was it really a healthy sample or should I test it with PCR nonetheless to be sure?) instead of being misleading with high confidences that could be overfitting the training data. We stressed this in the new Discussion paragraph as well.
Q2: Many parameters in Table 2 have overly complex and lengthy subscripts. We recommend the authors simplify them. For example: λ540:580.
A2: We removed the subscripts as requested.

Reviewer 3 Report
Comments and Suggestions for Authors
This work aims to develop a framework for the early detection of Flavescence dorée (FD) in grapevine leaves using hyperspectral imaging and vegetation indices.The study is good and includes a valuable scientific contribution; however, there are some concerns and issues that the esteemed authors should be made aware of and address in order for the paper to be suitable for publication.
1. The number of samples (201 total) appears small for hyperspectral classification, especially when split into three classes and further divided into training and testing subsets.
The statistical power and robustness of the model are therefore questionable.
2. The “balanced per-class” statement is unclear. Table 1 suggests balanced totals (67 per class), but it is not clarified whether balance was achieved by random undersampling or oversampling. Clarify balancing methodology and whether any synthetic balancing was done before ML training.
3. The acquisition under natural indoor light (“facility with a window from 10 AM to 12 AM”) introduces uncontrolled illumination variability. The absence of artificial light standardization or calibration lamps severely limits the reliability of reflectance spectra. Discuss the possible spectral distortion due to variable sunlight angles and intensity, and describe any normalization or correction measures beyond simple Teflon referencing.
4. The acquisition delay of ~15 minutes after sampling could lead to leaf dehydration and altered reflectance spectra. The authors should justify how this delay was standardized or mitigated across all samples.
5. The Field of View (FoV) and camera-object distance are clearly stated, but camera calibration (radiometric and geometric) is not discussed. Without this, reproducibility is compromised.
6. The resizing of hyperspectral cubes from 1024×1080 to 640×480 px for computational convenience introduces potential spectral-spatial distortion. There is no justification for the chosen resizing method (nearest neighbor, bilinear, etc.) or evaluation of the loss of spectral integrity. Provide a validation that resizing did not alter the spectral signatures used in ML.
7. The use of Roboflow AI segmentation is an interesting approach but insufficiently validated. There is no reference to segmentation accuracy, nor is there evidence that masks were visually inspected or cross-checked. Add performance metrics for segmentation or show a few representative segmentation results in supplementary figures.
8. The outlier removal criterion (“values above the 75% quartile or below 25%”) is conceptually incorrect. Quartiles define interquartile ranges (IQR), not outlier thresholds. Correct the description or provide the actual method (e.g., using 1.5×IQR rule).
9. The use of quartile values and mean as summary statistics for vegetation indices is a strong dimensionality reduction choice but lacks justification. Explain why higher-order moments (variance, skewness) were not considered. Discuss whether this summary is sufficient to capture within-leaf heterogeneity.
10. The selection of only five basic models (Decision Tree, Naive Bayes, k-NN, Ensemble Trees, and Shallow NN) is rather simplistic for hyperspectral data. The omission of SVM or deep CNNs is justified only by a vague statement about “negative outputs,” which is scientifically weak. Most SVM implementations in MATLAB handle probabilistic outputs properly; hence, exclusion should be empirically, not theoretically, justified.
11. There is no mention of hyperparameter tuning, feature importance analysis, or performance metrics (accuracy, F1, confusion matrix) for model evaluation. These are critical for reproducibility.
12. Cross-validation was mentioned (“5 folds”) but not clearly linked to the Dtrain/Dtest split. Clarify whether cross-validation was applied only to Dtrain or across the whole dataset.
13. The approach of generating synthetic samples using Monte Carlo methods is innovative but poorly justified.
14. There is no clear validation that synthetic samples faithfully represent real spectral variability.
15. Numerous grammatical inconsistencies, typographical errors (e.g., “acquistions,” “RAM,and500 GBofstorage”), and formatting issues make reading difficult. A thorough language revision by a native English editor is required.
Overall the paper is good and the presentation of the results is very good, please take into consideration the comments I mentioned earlier. Thank you very much
Author Response
Q1. The number of samples (201 total) appears small for hyperspectral classification, especially when split into three classes and further divided into training and testing subsets. The statistical power and robustness of the model are therefore questionable.
A1: We thank the reviewer for this comment. We do agree that 201 hypercubes only is a small dataset, especially considering that we have only 47 hypercubes per class in Dtrain. This is exactly the reason why we developed a complicated uncertainty estimation procedure as detailed in Section 2.5 onward. There, we also stated that the dataset size is an issue and for this reason computed predictors variability of both original and synthetic data (Section 2.5.1, σ1 and σ2). Fig.4 in the Results Sections highlights how the two variabilities are comparable and thus validates the adoption of the Monte Carlo augmenting approach. In addition, we wish to stress that the aim of this work was to demonstrate the applicability of the uncertainty-aware method that builds upon existing ML models by aggregating their standard outcomes with probability values coming from z-testing as detailed in Section 2.5.2. For this reason, even if we do agree that extensive validation with new unseen data is necessary in the future, we still believe our contribution to be valuable at least as a demonstration of applicability. We stressed this matter in Sections 2.4 and 2.5 and discussed it again in the “Limitation of the study” section.
Q2: The “balanced per-class” statement is unclear. Table 1 suggests balanced totals (67 per class), but it is not clarified whether balance was achieved by random undersampling or oversampling. Clarify balancing methodology and whether any synthetic balancing was done before ML training.
A2: Class balancing here refers simply to the fact that an equal number of samples belonging to each class were collected. This means that after receiving the outcomes of PCR tests on the samples after each collection campaign, we kept only those resulting positive and built the asymptomatic class samples according to the procedure described in Section 2.2. Healthy samples were included in the corresponding healthy class if they resulted negative without any other condition. This procedure was done manually in collaboration with expert agronomists before analyzing the hypercubes (so just by checking the result of PCR analysis). To better clarify this step, we improved Section 2.2.
Q3. The acquisition under natural indoor light (“facility with a window from 10 AM to 12 AM”) introduces uncontrolled illumination variability. The absence of artificial light standardization or calibration lamps severely limits the reliability of reflectance spectra. Discuss the possible spectral distortion due to variable sunlight angles and intensity, and describe any normalization or correction measures beyond simple Teflon referencing.
A3: We understand the reviewer’s concern. As mentioned by the camera’s manufactures, Spectralon referencing and white balancing were fundamental steps to remove spectral distortions. Moreover, most of these distortions appear in the infrared region (wavelengths higher than 800 nm), which were not included in the Vis formulations and thus do not affect the experiments. We mentioned this in the modified text of Section 2.3.2.
Q4: The acquisition delay of ~15 minutes after sampling could lead to leaf dehydration and altered reflectance spectra. The authors should justify how this delay was standardized or mitigated across all samples.
A4: We understand the reviewer’s concern, however it is worth noting that right after collection the leaves were placed in a gardener sponge filled with water to prevent dehydration. In any case, vine leaves have moderate respiration rates allowing them to avoid dehydration for several hours even if not stored in the fridge, as supported by https://doi.org/10.1016/j.scienta.2021.110627 . We added this comment in Section 2.2.
Q5: The Field of View (FoV) and camera-object distance are clearly stated, but camera calibration (radiometric and geometric) is not discussed. Without this, reproducibility is compromised.
A5: Radiometric calibration of the camera (HERA VIS-NIR produced by NIREOS, patented technology) is performed by the proprietary software at startup since the sensing technology is based on Fourier Transform. Mainly, the procedure requires the user to select data processing parameters related to the signal processing pipeline to reduce the acquisition noise and select the best integration time for the hypercube acquisition. To improve results, the manufacturer suggests to perform intensity calibration using Spectralon when conducting measurements, as we did. This is why in our work we only used normalized reflectance that is obtained after intensity correction. Geometric calibration was not needed since a standard calibration information was already included in the proprietary software used for the acquisitions. We added this information in Section 2.1.
Q6: The resizing of hyperspectral cubes from 1024×1080 to 640×480 px for computational convenience introduces potential spectral-spatial distortion. There is no justification for the chosen resizing method (nearest neighbor, bilinear, etc.) or evaluation of the loss of spectral integrity. Provide a validation that resizing did not alter the spectral signatures used in ML.
A6: We thank the reviewer for this interesting question. The chosen method for interpolation used by the resizing method is “bicubic”, which produces as output a weighted average of values in the nearest 4x4 neighborhood. However, the resizing was done per wavelength; meaning that given the 2D matrix corresponding to a specific wavelength of the original hypercube, we applied the resizing method so that interpolation issues that may arise mostly affect spatial integrity instead of spectral integrity. As proof of the validity of our approach, in the graph below we show the per-band values of “Diff”, calculated per leaf sample as the average original spectra minus the average spectra of the corresponding resized leaf. Values in the barplot below are the mean values of such differences. As the reviewer can observe, the maximum differences appear in the infrared region reaching a 0.0025 difference among normalized reflectance values (dimensionless) with standard deviation of +- 0.0008 (dimensionless). This shows that our resizing method is valid and does not add any unwanted noise or distortion in the data. We modified Section 2.3.1 accordingly with details about the interpolation and the maximum difference observed. For brevity reasons, we did not include the graph below.

Q7: The use of Roboflow AI segmentation is an interesting approach but insufficiently validated. There is no reference to segmentation accuracy, nor is there evidence that masks were visually inspected or cross-checked. Add performance metrics for segmentation or show a few representative segmentation results in supplementary figures.
A7: In Roboflow AI the process to generate masks is as follows:
- Load the images you wish to label
- Start a labeling session and, for each image, click on the object inside it that you wish to segment. In our case, we manually selected the leaf in each RGB picture. Segmentation this way is easy since leaves were always depicted over a white background so the algorithm had almost a 100% accuracy rate.
- If the segmentation mask was incorrect, modify it by moving around the contour points
We then converted the generated labels from Roboflow AI format to a binary mask using a custom-made MATLAB software we developed.
As the reviewer can see, the process is very simple and it’s almost impossible for Roboflow AI to produce wrong masks. Even in the rare cases this happened, each mask was visually inspected and corrected by the authors. Roboflow AI was simply used as a platform to speed up the process of mask generation but was supervised by a user in every step of the procedure, so no mask was generated without the authors checking it. As a consequence, performance metrics not only cannot be obtained (the platform does not provide them since obviously to get a performance you must have a ground truth, but the ground truth in this case is the label we wish to obtain), but even if they could be obtained they would be useless. We modified Section 2.3.1 accordingly to better clarify the procedure. Please refer to our previous works (refs 30-31) for specific pictures of RGB samples of the leaves and binary masks.
Q8: The outlier removal criterion (“values above the 75% quartile or below 25%”) is conceptually incorrect. Quartiles define interquartile ranges (IQR), not outlier thresholds. Correct the description or provide the actual method (e.g., using 1.5×IQR rule).
A8: We checked our code to be sure of the actual method adopted and found out we used MATLAB’s “isoutlier” function in “quartiles” mode, which applies the method stated by the reviewer (removing data outside the IQRx1.5 range). We revised section 2.3.2 accordingly.
Q9: The use of quartile values and mean as summary statistics for vegetation indices is a strong dimensionality reduction choice but lacks justification. Explain why higher-order moments (variance, skewness) were not considered. Discuss whether this summary is sufficient to capture within-leaf heterogeneity.
A9: Since FD symptoms may appear in the form of red spots of different shape, size and position on the leaf’s surface, we wanted to roughly represent this phenomenon by using statistical indicators. Ideally, to healthy leaves of uniform color correspond distributions with low variance and small quartile ranges, while for leaves with red spots variance increases for those VIs that better capture the presence of red spots (like those linked to anthocyanins content). We wanted to keep the number of predictors to a reasonable number, so we selected only a subset of statistical operators that can roughly represent the VIs distributions in the leaf sample. Moreover, we used other statistical parameters such as skewness and kurtosis in our previous works at refs 30-31, but they were the least informative ones so we decided to remove them. This allows us to keep a reasonable number of predictors without reducing the overall performance of the models too much. We added this discussion in Section “Limitations of the study”.
Q10: The selection of only five basic models (Decision Tree, Naive Bayes, k-NN, Ensemble Trees, and Shallow NN) is rather simplistic for hyperspectral data. The omission of SVM or deep CNNs is justified only by a vague statement about “negative outputs,” which is scientifically weak. Most SVM implementations in MATLAB handle probabilistic outputs properly; hence, exclusion should be empirically, not theoretically, justified.
A10: As reflected by the article title and by several statements throughout the article, the focus of the article was not to test every possible ML model but, instead, to demonstrate the applicability of the uncertainty-aware structure we proposed that builds upon existing ML models. As a result, we limited the choice to the off-the-shelf models available in MATLAB, one per category, that output a probability value. SVMs and Kernels were excluded because the output is not a probability but instead a value representing the support vector/kernel which, if mapped by the model into the target classes, produces a category. To apply our method, it is necessary to have a probability measure (the confidence score) as detailed in Section 2.5.2. CNNs were not tested since these kinds of models not only are computationally heavy to train but also perform better with matricial datasets. The closest architecture is the WNN tested and named ML5. In future developments we will test CNNs as well. We added this information in Section 2.4 and in the “Limitations of the study” section.
Q11: There is no mention of hyperparameter tuning, feature importance analysis, or performance metrics (accuracy, F1, confusion matrix) for model evaluation. These are critical for reproducibility.
A11: Hyperparameters of choice were already mentioned per model in Section 2.4 according to MATLAB’s tunable parameters. We stress again that we used the standard models available on the platform since the goal was not to produce the best ML model but to provide a demonstration of the applicability of our uncertainty-aware methodology. Models’ performance metrics were added in the new Table 4. Feature importance was conducted using Shapley importance approaches and was used to compute uncertainty-aware probabilities, as mentioned in Section 2.5.2 already. Showing the specific values of the Shapley importance analysis is not relevant for the purpose of this article, hence they were omitted.
Q12: Cross-validation was mentioned (“5 folds”) but not clearly linked to the Dtrain/Dtest split. Clarify whether cross-validation was applied only to Dtrain or across the whole dataset.
A12: Cross-validation is computed by MATLAB’s after training on the data in Dtrain. We revised Section 2.4 accordingly to clarify this matter further.
Q13: The approach of generating synthetic samples using Monte Carlo methods is innovative but poorly justified.
A13: We revised Section 2.5 to clarify the rationale and also provide a summary of the method. To summarize, we faced two issues: (1) the low number of data samples in our dataset, insufficient to calculate the input data uncertainty, and (2) the impossibility to propagate this uncertainty inside the model structure, as it would have required changes of the models’ structure accordingly. To address issue (1) we first generated synthetic hypercubes following the details in Section 2.5.1. The aim of this step was to artificially generate uncertainty of the input data, named σ2. Values were then compared with the original data variability σ1 to assess their validity (see Fig. 4). Then, we artificially generated feature predictors using σ2 as the uncertainty parameter for the second Monte Carlo sampling approach. Issue number (2) was addressed by simply aggregating the confidence scores obtained as output from the models with values obtained by z-testing the generated data to check the probability of them belonging to a certain class (feature-based). This allows us to generalize the uncertainty-awareness without changing the models’ structures as the aggregation happens afterward.
Q14: There is no clear validation that synthetic samples faithfully represent real spectral variability.
A14: We kindly refer the reviewer to Fig. 4, showing the variability of original predictors σ1 (calculated from the data in Dtest) and the variability of synthetic predictors obtained after generating the synthetic hypercubes (see Section 2.5.1), σ2. The two variabilities are shown per class and per predictor and are comparable in all cases, highlighting that synthetic generation is a valid method that properly represents the original phenomena. Further validations should be made considering new data taken from other varieties as well, as mentioned in the “Limitations of the study” section.
Q15: Numerous grammatical inconsistencies, typographical errors (e.g., “acquistions,” “RAM,and500 GBofstorage”), and formatting issues make reading difficult. A thorough language revision by a native English editor is required.
A15: We revised the document and corrected the typos we found.

Reviewer 4 Report
Comments and Suggestions for Authors
This paper presents a methodologically interesting approach introducing uncertainty-aware models for early disease detection in viticulture using hyperspectral data. The concept addresses an important gap in current precision agriculture imaging—quantifying uncertainty in classification outcomes. However, while technically strong, the work remains primarily a proof-of-concept without robust biological validation.
Major Comments
1. The model’s robustness should be tested on external vineyard data (different year or variety). Synthetic augmentation cannot fully replace empirical variability.
2. Currently, the “uncertainty” is purely statistical. Include field repeatability experiments to demonstrate correspondence between model uncertainty and spectral variance.
3.Clarify novelty over previous works [30, 31]. As this is an extended version, explicitly differentiate which findings are new beyond data doubling and per-pixel features.
4. What specific VIs or spectral ranges best signal asymptomatic FD infection? Include biological interpretation of high-weight predictors (e.g., anthocyanin indices mARI/mARI2).
5. Section 3 contains dense mathematical detail. A schematic overview with parameter values and computational cost would enhance readability.
6. Discuss how this approach could integrate with UAV or proximal sensing platforms—currently the study is lab-based.
Minor Comments
1. Improve English grammar and streamline long sentences.
2. Add full axis labels, legends, and better color contrast in Figures 4–5.
3. Include a short limitations paragraph in the conclusion.
4. Clarify whether Monte Carlo generation introduces label leakage (same leaf used multiple times with perturbation).
Author Response
Q1: The model’s robustness should be tested on external vineyard data (different year or variety). Synthetic augmentation cannot fully replace empirical variability.
A1: We added a comment about this in the new “Limitation of the study” section.
Q2: Currently, the “uncertainty” is purely statistical. Include field repeatability experiments to demonstrate correspondence between model uncertainty and spectral variance.
A2: The uncertainty shown in this work refers to a combination of population uncertainty calculated by z-test (see Section 2.5.2) and the classification confidence score. Hence, it describes the measurement pipeline uncertainty as a whole (from acquisition to classification). A repeatability experiment was conducted as described in Section 2.5.1 to compute the spectral variability of the samples (see the start of Section 2.5.1). We do agree that any conclusion about FD incidence must be validated with new data collected from the field, but this is a slow process requiring collaborations with commercial vineyards and a whole growing season. The aim of the article was not to draw conclusions on the biological aspects of the disease, but instead to demonstrate how metrological uncertainty can be added to the measurement pipeline to provide useful guidelines and insights about detection of asymptomatic samples. We added a comment about this aspect in the new “Limitation of the study” section.
Q3: Clarify novelty over previous works [30, 31]. As this is an extended version, explicitly differentiate which findings are new beyond data doubling and per-pixel features.
A3: Technical extensions were already specified at lines L#109-116. The reader can easily verify that from section 2.5 onward the presented article is completely novel compared to previous works (referring to the uncertainty-aware models, which are the core of the article). We revised lines L#113-116 to better highlight the novelty of this aspect.
Q4: What specific VIs or spectral ranges best signal asymptomatic FD infection? Include biological interpretation of high-weight predictors (e.g., anthocyanin indices mARI/mARI2).
A4: We thank the reviewer for this comment. However, this aspect was not relevant for the present article and was already explored in our previous works. We briefly mentioned this in the new “Discussion” section.
Q5: Section 3 contains dense mathematical detail. A schematic overview with parameter values and computational cost would enhance readability.
A5: Referring to the new Section 2.5 (old Section 3 in the original article), we modified the text to better summarize the whole procedure described in detail in the following Sections 2.5.1 and 2.5.2. We added Table 3 summarizing relevant parameters.
A6: Discuss how this approach could integrate with UAV or proximal sensing platforms—currently the study is lab-based.
Q6: As the reviewer noticed, the camera used requires several seconds to capture a hypercube and its density is more suited for detailed studies. Despite the study was not conducted in a laboratory, we agree that a different technology should be adopted for real-time measurements. The present study should be considered as a stepping stone providing insights useful for FD research so that only relevant VIs and corresponding predictors could be selected and applied to less dense spectra acquired by multispectral cameras mounted on UAVs or land rovers. We added this discussion in the “Limitations of the study” section.
Q7: Minor Comments. (1) Improve English grammar and streamline long sentences. (2) Add full axis labels, legends, and better color contrast in Figures 4–5. (3) Include a short limitations paragraph in the conclusion. (4) Clarify whether Monte Carlo generation introduces label leakage (same leaf used multiple times with perturbation).
A7: (1) We checked the article and corrected all the English typos and long sentences we found. (2) Full axis labels and legends are already present in both figures. Colors were chosen by checking if the images printed in grayscale were understandable enough, so we do not think they need to be modified. (3) We added a “Limitations of the study” section before the Conclusions. (4) Label leakage refers to the contamination of the model input features with outcome information. However, models were trained on Dtrain and tested on Dgaussian. Data in Dgaussian was generated from Dtest, a subdataset that did not include any data from Dtrain. So, all samples were unseen by the models. Maybe the reviewer’s concern refers to the synthetic leaf generation (Dsynth) as explained in Section 2.5.1, for which 500 new synthetic samples were generated starting from each sample in Dtest. However this procedure was only necessary to compute input data uncertainty σ2 and were not used for the models’ testing. Instead, the computed σ2 was used as input of a new Monte Carlo process that generates predictors directly from the data in Dtest. Some generated samples could have similar values, however please consider that each feature of the new sample was generated independently using values in σ2 (different for each class), so this chance is relatively low. Also, generated samples in Dgaussian were never seen by the models (trained on Dtrain).

Reviewer 5 Report
Comments and Suggestions for Authors
In order to improve the quality of this work, some comments have been given as below.
1. Please clarify the innovation contribution, explicitly contrast this uncertainty-aware approach with prior works on uncertainty modeling in agricultural ML applications.
2. Authors are suggested to highlight the research gap in introduction.
3. Methodology section should be simplified or visualized. Sub-sections 3.1–3.2 are conceptually strong but overly technical. A schematic overview or pseudocode would aid comprehension.
4. Dataset limitations has been addressed. Please provide a discussion on the possible bias due to vineyard location, grape variety, or sampling year.
5. Please add statistical metrics. Beyond confusion matrices, authors should include precision, recall, F1-score, and AUC-ROC to allow a more comprehensive evaluation.
6. The depth of discussion should be improved. And authors should link findings to broader smart agriculture or precision viticulture frameworks, highlighting managerial implications.
7. Authors are suggested to expand uncertainty interpretation. Please explain what “unclassified” cases mean for practitioners and how such uncertainty could guide future inspection or testing.
Author Response
Q1: Please clarify the innovation contribution, explicitly contrast this uncertainty-aware approach with prior works on uncertainty modeling in agricultural ML applications.
A1: We thank the reviewer for this comment. We improved the Introduction Section stressing out that no other works dealt with uncertainty modeling in agricultural ML applications.
Q2: Authors are suggested to highlight the research gap in introduction.
A2: The modified version of the Introduction Section now highlights better the research gap and how this work tries to tackle it.
Q3: Methodology section should be simplified or visualized. Sub-sections 3.1–3.2 are conceptually strong but overly technical. A schematic overview or pseudocode would aid comprehension.
A3: We understand the difficulties while reading these Sections. We modified them (now Section 2.5 onward) providing a summary at the start of Section 2.5. Schemes of specific procedures were already provided in Fig. 2 and Fig. 3.
Q4: Dataset limitations has been addressed. Please provide a discussion on the possible bias due to vineyard location, grape variety, or sampling year.
A4: We address this issue in the new “Limitations of the Study” Section as well as discussing the limited size of the dataset.
Q5. Please add statistical metrics. Beyond confusion matrices, authors should include precision, recall, F1-score, and AUC-ROC to allow a more comprehensive evaluation.
A5: AUC-ROC curves are usually not informative for standard models, while they are a fundamental tool to understand the models’ behaviour while developing a new structure. As such, we prefer to keep things simple and avoid showing them in the paper. Moreover, these two curves provide similar information to precision, recall and F1-Score, so we prefer to show these parameters instead to save space (now shown in Table 4).
Q6: The depth of discussion should be improved. And authors should link findings to broader smart agriculture or precision viticulture frameworks, highlighting managerial implications.
A6: We added the new “Discussion” and “Limitations of the Study” Sections to deepen the discussion as requested. As for managerial implications, we think its too early to make any claim since only one variety was tested and the experimentation still relies on long acquisition and processing times compared to fast acquisitions provided by multispectral cameras mounted on land rovers or UAVs. We addressed this briefly in the “Limitations of the Study” Section.
Q7: Authors are suggested to expand uncertainty interpretation. Please explain what “unclassified” cases mean for practitioners and how such uncertainty could guide future inspection or testing.
A7: “Unclassified” means that the sample under test could potentially belong to multiple classes, so a more refined test should be conducted to properly determine if FD was present or not (e.g., PCR). We explained this concept bettere in the modified version of the Conclusions and in the new “Discussion” Section.
Reviewer 6 Report
Comments and Suggestions for Authors
The manuscript presents an innovative approach for early detection of Flavescence dorée (FD) symptoms through the use of hyperspectral imaging combined with machine learning models explicitly incorporating uncertainty estimation. The authors propose a Monte Carlo-based methodology to generate synthetic samples and to propagate uncertainty through classification models, introducing the concept of “uncertainty-aware” decision systems. The topic is timely and relevant for the precision agriculture and plant-disease-monitoring communities. The work is generally well organized and scientifically sound, and the presentation demonstrates strong technical competence.
Nevertheless, the manuscript would benefit from substantial revision to improve clarity, conciseness, and scientific interpretation. The main weaknesses concern (i) redundancy and length in the introduction, (ii) limited biological interpretation of the results, (iii) insufficient discussion of methodological assumptions, and (iv) the quality and readability of figures and tables. The inclusion of quantitative performance metrics beyond overall accuracy would also strengthen the study’s credibility.
Major Comments
- The novelty lies in introducing uncertainty quantification within hyperspectral machine learning applied to FD detection. However, the current text tends to dilute this contribution in an overly extended literature review. The introduction should be condensed and explicitly state what methodological gap the proposed framework fills, how it differs from previous FD classification works, and what its implications are for field diagnosis.
- Several methodological choices appear arbitrary or lack justification. Examples include the selection of ten vegetation indices, the use of four summary statistics (quartiles and mean) as predictors, and the decision to generate 500 Monte Carlo realizations per sample. Each of these should be justified in terms of statistical representativeness or prior work. Moreover, the rationale for using synthetic data instead of data augmentation or cross-validation should be better explained.
- The results rely mainly on classification accuracy. Standard performance measures such as precision, recall, specificity, and F1-score should be included to allow readers to assess model robustness and to compare results with related studies. In addition, the interpretation of the Bayesian test would be clearer if the probabilities were complemented with confidence intervals or variance estimates.
- The discussion focuses heavily on numerical comparison between models, but gives limited attention to biological and physical interpretation. For instance, the manuscript mentions that variations in mARI and mARI2 are linked to anthocyanin content and disease stress; this point could be developed further to highlight physiological insights. Similarly, the relationship between uncertainty distributions and the optical properties of infected leaves could be discussed.
- Figures are informative but dense, mostly Figure 3. Figure 5 combines five models, confusion matrices, and trend graphs in one page, which makes it visually overloaded; splitting the panels or moving supplementary details to an appendix would improve readability. Table 2 should be formatted using consistent mathematical notation, with clear indication of wavelength ranges and units.
Section-by-Section Comments
- The abstract is coherent and informative, but should be more quantitative. It is recommended to report the best model explicitly, the achieved accuracy per class, and the key result regarding uncertainty reduction. The current phrasing “the best models belonged to the class of decision trees” should be replaced by a precise statement such as “the ensemble of bagged decision trees achieved 75.7% accuracy for asymptomatic samples, outperforming the baseline model by 26 percentage points”.
- The introduction provides a thorough review of FD detection methods and hyperspectral imaging. However, several paragraphs (lines 25–110) repeat background information and could be merged. The final paragraph should clearly articulate the research gap, hypotheses, and main objectives in a concise manner. A short statement outlining how uncertainty quantification improves the diagnostic reliability compared to previous machine learning studies would enhance the narrative focus.
- The technical description of data acquisition and preprocessing is detailed and reproducible. Nonetheless, the following aspects require clarification:
- Explain why only the 400–1000 nm range was used when previous studies also emphasize 1600–2200 nm as informative for FD (Obviously, it will be related to the sensor used, but it is important to address this matter.).
- Justify the use of quartiles and mean instead of other statistical descriptors such as standard deviation or skewness.
- Specify whether Z-score normalization was applied globally or per class, as this affects the comparability of predictors.
- Provide the exact number of pixels analyzed per leaf and whether any data-balancing strategies were used beyond equal class sizes.
- Discuss the potential impact of natural illumination variability on the reflectance values and how it was controlled.
- This section is conceptually interesting but mathematically dense. The explanation of the probability aggregation (Pm, Zm, Um) would be clearer with an intuitive summary preceding the equations. The statistical meaning of the acceptance threshold t and its implications for classification outcomes should be described in practical terms. Equation (1) appears complex and may be unfamiliar to many readers; providing a brief derivation or reference justification would be helpful.
- The quantitative results are convincing and show that decision-tree-based models outperform other classifiers. However, the discussion could be enriched by connecting numerical findings to biological meaning. For example, the high variability of anthocyanin-related indices (mARI, mARI2) might reflect the pigment redistribution in early FD stages, and this physiological explanation would strengthen the argument for the proposed indicators. In addition, it would be useful to compare the obtained accuracy with previously published hyperspectral or UAV-based studies to contextualize the performance.
- The conclusions are adequate but could be more concise and forward-looking. It is recommended to explicitly state the main achievement (integration of uncertainty into FD classification) and its practical implication (improved reliability for early detection). The final paragraph discussing future work could be shortened to focus on two concrete directions: (i) extension to deep learning frameworks, and (ii) validation across different grape varieties.
Minor Issues
- The citation style should be carefully checked for consistency with MDPI format (spacing, punctuation, and author-year sequence).
- Abbreviations such as FD, HSI, VI, and ML should be defined once at first occurrence only.
- Verify that all units are expressed in SI format (e.g., “5 nm spectral resolution” rather than “5nm”).
- Ensure that all references are recent and relevant; several could be omitted or merged to streamline the bibliography.
Author Response
Q1: The novelty lies in introducing uncertainty quantification within hyperspectral machine learning applied to FD detection. However, the current text tends to dilute this contribution in an overly extended literature review. The introduction should be condensed and explicitly state what methodological gap the proposed framework fills, how it differs from previous FD classification works, and what its implications are for field diagnosis.
A1: Thank you for this comment. We shortened the introduction section as requested and made the novelty of our work the central focus.
Q2: Several methodological choices appear arbitrary or lack justification. Examples include the selection of ten vegetation indices, the use of four summary statistics (quartiles and mean) as predictors, and the decision to generate 500 Monte Carlo realizations per sample. Each of these should be justified in terms of statistical representativeness or prior work. Moreover, the rationale for using synthetic data instead of data augmentation or cross-validation should be better explained.
A2: The selection of the 10 VIs described in Section 2.3.2 and Table 2 was made according to existing literature (see Table 2), background research (see the Introduction) and previous work (see refs. 30-31). We made this choice clearer at the start of Section 2.3.2 and in the “Discussion” Section as requested by the reviewer. The summary statistics chosen was a design choice backed up by the fact that the distribution of VIs corresponding to each pixel of the leaf can be statistically represented by statistical values such as quartiles and means. Their effectiveness was already proven in our previous works and is validated by this work’s results (see Fig. 4). The 500 Monte Carlo simulations conducted per Dtest sample (totaling 30,000 samples) are justified by the lack of computational power to simulate more samples even on several workstations (we stress again that simulating a whole leaf of more than 200,000 pixels requires time) and by the “Guide to the Expression of Uncertainty” (ref 49) section 7.2 as already stated in Section 2.5.1. The rationale for using synthetic data instead of augmentation is described at the start of Section 2.5, now modified to better clarify the approach and the idea behind it. Synthetic data allow to apply Monte Carlo simulations and thus compute uncertainty. With our approach, we first analyzed the variability of predictors computed from synthetic leaves generated with Monte Carlo and assessed their validity by comparing them with the original ones (Section 2.5.1). Then, this allows us to directly generate predictors using the simulated variabilities σ2, thus taking into account the uncertainty of the input data this way. Augmentation of the data is equally demanding in terms of computational power and heavily depends on computer vision techniques instead of statistics, so the augmented data generated this way could have been completely different from the actual phenomena. Cross validation of the trained models was actually performed as stated in Section 2.4.
Q3: The results rely mainly on classification accuracy. Standard performance measures such as precision, recall, specificity, and F1-score should be included to allow readers to assess model robustness and to compare results with related studies. In addition, the interpretation of the Bayesian test would be clearer if the probabilities were complemented with confidence intervals or variance estimates.
A3: Precision, recall, specificity and F1-scores are now provided in Table 4 for all models (original version vs uncertainty-aware version using the best acceptance threshold t). As for the request about the confidence intervals of values obtained from the Bayesian test, in our case this is simply reduced to showing the histograms of the predicted probabilities Um according to the original gtruth class and cutting the tails to get a 95% interval. Keep in mind that these distributions are not Gaussian. However, we believe that adding this information will inevitably increase the complexity of the article which is already quite dense. Hence, we prefer to keep it simple and consider the results of Section 2.5.3 in Table 5 as just another metric that complements the ones shown in Table 4 and Fig. 5 to understand the behaviour of models.
Q4: The discussion focuses heavily on numerical comparison between models, but gives limited attention to biological and physical interpretation. For instance, the manuscript mentions that variations in mARI and mARI2 are linked to anthocyanin content and disease stress; this point could be developed further to highlight physiological insights. Similarly, the relationship between uncertainty distributions and the optical properties of infected leaves could be discussed.
A4: Biological and physical interpretations are beyond the scope of the article, which was the demonstration of the applicability of our uncertainty-aware models to tackle complex phenomena like FD. Our previous works (refs. 30-31) already explored the effect of specific VIs and link them with the leaves’ physiological status, as well as the background literature cited in the Introduction Section. Alongside this topic, we also added a discussion about the relationship between uncertainty and optical properties in the new Discussion section (3.1).
Q5: Figures are informative but dense, mostly Figure 3. Figure 5 combines five models, confusion matrices, and trend graphs in one page, which makes it visually overloaded; splitting the panels or moving supplementary details to an appendix would improve readability. Table 2 should be formatted using consistent mathematical notation, with clear indication of wavelength ranges and units.
A5: We understand the concerns of the reviewer about Figure 5, however splitting it will reduce its effect since it must be read all together to allow quick comparison of results among models and the effect of the applied uncertainty-aware structure. As a result, Figure 5 is a central figure for discussing results and cannot be split nor moved to supplementary materials. As for Table 2, we changed the mathematical notation to make wavelength ranges clearer as requested.
Q6: The abstract is coherent and informative, but should be more quantitative. It is recommended to report the best model explicitly, the achieved accuracy per class, and the key result regarding uncertainty reduction. The current phrasing “the best models belonged to the class of decision trees” should be replaced by a precise statement such as “the ensemble of bagged decision trees achieved 75.7% accuracy for asymptomatic samples, outperforming the baseline model by 26 percentage points”.
A6: We modified the abstract as requested.
Q7: The introduction provides a thorough review of FD detection methods and hyperspectral imaging. However, several paragraphs (lines 25–110) repeat background information and could be merged. The final paragraph should clearly articulate the research gap, hypotheses, and main objectives in a concise manner. A short statement outlining how uncertainty quantification improves the diagnostic reliability compared to previous machine learning studies would enhance the narrative focus.
A7: Lines 25-110 outline the background information including the disease’s characteristics, economic impact on EU countries, current methods for diagnosis and research background to address it. Therefore, we think all the information contained in these lines is necessary to understand the context of our work. We modified this section to make it less verbose while keeping all the necessary information. As for the last paragraph, we edited it accordingly outlining the impact of uncertainty on the detection of asymptomatic FD samples.
Q8: Explain why only the 400–1000 nm range was used when previous studies also emphasize 1600–2200 nm as informative for FD (Obviously, it will be related to the sensor used, but it is important to address this matter.).
A8: The reason was technical: we did not have a sensor capable to get readings outside of the specified range of 400-1000 nm. Nonetheless, the majority of studies cited as background literature show that a lot of information is already contained in the range 400-1000 while only a few interesting properties can be obtained in the range 1600-2200 (mostly affected by scattering effects). We agree that including this range can be an additional information useful to better characterize the phytoplasma symptoms, hence we mention it in the new “Limitations of the study” Section (Section 3.2).
Q9: Justify the use of quartiles and mean instead of other statistical descriptors such as standard deviation or skewness.
A9: In our previous work (ref. 31) we used also outliers, the position of the boxplot whiskers, kurtosis and skewness alongside the other 4 predictors we used in this work, but noticed that their impact was relatively low compared to the 4 we chose. Hence, we decided to discard them to keep the number of predictors to a reasonable number (4 predictors x 10 VIs) without losing prediction capacity. We added this motivation in the “Discussion” section to clarify our choice.
Q10: Specify whether Z-score normalization was applied globally or per class, as this affects the comparability of predictors.
A10: Z-score normalization was applied globally. This information was clarified better in the text in Section 2.3.2.
Q11: Provide the exact number of pixels analyzed per leaf and whether any data-balancing strategies were used beyond equal class sizes.
A11: All leaves in our dataset are different; as a result, the number of pixels that represent them are different. We already provided a pixel range in the text (100k – 300k depending on the leaf’s size). This information is now contained in the new Table 3 instead. Other data-balancing techniques are not needed since we (1) collected the same amount of samples per class, and (2) compute the leaf’s statistics so that only a statistical summary of the leaf’s VIs values is reported in the final dataset (one row per sample), thus reducing the impact of the number of pixels in the leaf. This is already explained in the text in Section 2.3.2, nonetheless we slightly modified this Section to make our point clearer.
Q12: Discuss the potential impact of natural illumination variability on the reflectance values and how it was controlled.
A12: The variability of natural illumination mostly impacts the light scattering effects appearing in range > 800 nm. These effects are kept under control since we adopted VIs that consider a span of wavelengths and compute means averaging these values with other ones captured at lower wavelengths. The range > 800 nm was not used by the VIs considered in our work (see Table 2 for their formulation). We added this information in Section 2.3.2.
Q13: This section is conceptually interesting but mathematically dense. The explanation of the probability aggregation (Pm, Zm, Um) would be clearer with an intuitive summary preceding the equations. The statistical meaning of the acceptance threshold t and its implications for classification outcomes should be described in practical terms. Equation (1) appears complex and may be unfamiliar to many readers; providing a brief derivation or reference justification would be helpful.
A13: We added a summary of our approach at the start of Section 2.5. Also, Section 2.5.2 must be read while carefully looking at Figure 3 which describes the mathematical steps with understandable figures (values shown in the figure are not random, by applying the formulas the reader can obtain them and cross-check results). Reference justifications were already provided in the text preceding the equations at refs 47, 48, 50. The acceptance threshold is explained in Section 2.5.2 already and does not need further clarification. A practical example is already shown in the results section in Figure 5 (look at the graphs demonstrating the impact on the resulting accuracy by increasing t).
Q14: The quantitative results are convincing and show that decision-tree-based models outperform other classifiers. However, the discussion could be enriched by connecting numerical findings to biological meaning. For example, the high variability of anthocyanin-related indices (mARI, mARI2) might reflect the pigment redistribution in early FD stages, and this physiological explanation would strengthen the argument for the proposed indicators. In addition, it would be useful to compare the obtained accuracy with previously published hyperspectral or UAV-based studies to contextualize the performance.
A14: A link between Vis values and the samples’ class was already discussed in our previous works (refs 30-31) and is mentioned again in the new “Discussion” section. Comparing our results with other articles is not the focus of our work since our uncertainty-aware approach is applied on top of existing classifiers, as a result any ML architecture is suitable for our method (provided that the outputs are confidence scores). Moreover, UAV-based studies mainly adopt multispectral cameras or low-resolution hyperspectral cameras to properly conduct on-the-go fast measurements in the field, in contrast with our acquisitions that lasted a few seconds (due to the high density of the acquired hypercube). For these reasons, we think their performance is not comparable with our approach for now. UAV-related works were cited in the Introduction section already providing context for our study. We discussed this matter in the new “Limitations of the study” Section.
Q15: The conclusions are adequate but could be more concise and forward-looking. It is recommended to explicitly state the main achievement (integration of uncertainty into FD classification) and its practical implication (improved reliability for early detection). The final paragraph discussing future work could be shortened to focus on two concrete directions: (i) extension to deep learning frameworks, and (ii) validation across different grape varieties.
A15: We thank the reviewer for this suggestion. We modified the Conclusions section accordingly.
Q16: Minor Issues: (1) The citation style should be carefully checked for consistency with MDPI format (spacing, punctuation, and author-year sequence). (2) Abbreviations such as FD, HSI, VI, and ML should be defined once at first occurrence only. (3) Verify that all units are expressed in SI format (e.g., “5 nm spectral resolution” rather than “5nm”). (4) Ensure that all references are recent and relevant; several could be omitted or merged to streamline the bibliography.
A16: We thank the reviewer for these suggestions. We addressed all the minor issues accordingly by carefully revising the text.
Round 2
Reviewer 1 Report
Comments and Suggestions for Authors
The authors have revised the manuscript, but still the following modifications are not properly addressed.
Q8: Separate the results and discussion sections.
Currently results section is too short, this confuses the manuscript structure.
Q10: Show the real reflectance for each class.
A10: According to the camera’s manufacturer guidelines (HERA VIS-NIR produced by NIREOS), intensity calibration of the camera must be conducted every time the camera is used to properly adjust the measured quantities according to the experimental set-up conditions, because the measurement technology is based on Fourier Transform. This is why in our work we only used normalized reflectance that is obtained after intensity correction. Thus, the “real reflectance” the reviewer is referring to (which we think it is the absolute reflectance measured without normalization) cannot be used since it’s not reliable for this specific instrument.
Need clarification: It is difficult to understand that why real reflectance cannot be shown here?, A comparison graph can be made.
Q13: Figure 5. Move it to the results section.
Q14: Shorten the conclusion section.
It is still too long.
In addition: Add real pictures of the samples and imaging setups.
Author Response
Q1: Separate the results and discussion sections. Currently results section is too short, this confuses the manuscript structure.
A1: We partially moved some details from the Discussion subsection to the main Results section to limit this effect. Please note that the “Discussion” and “Limitations of the study” subsections belong to the Results sections, so the overall structure of the manuscript is consistent.
Q2: Show the real reflectance for each class.
(Previous answer): According to the camera’s manufacturer guidelines (HERA VIS-NIR produced by NIREOS), intensity calibration of the camera must be conducted every time the camera is used to properly adjust the measured quantities according to the experimental set-up conditions, because the measurement technology is based on Fourier Transform. This is why in our work we only used normalized reflectance that is obtained after intensity correction. Thus, the “real reflectance” the reviewer is referring to (which we think it is the absolute reflectance measured without normalization) cannot be used since it’s not reliable for this specific instrument.
(New question): Need clarification: It is difficult to understand that why real reflectance cannot be shown here? A comparison graph can be made.
A2: If the reviewer is referring to the “absolute reflectance” values, we unfortunately cannot show this data because we only saved the normalized hypercubes (containing normalized reflectance, as mentioned in the previous answer and in the article). As it is suggested also by refs. 33-34, normalizing the hypercube allows minimization of non-biological factors such as leaf tilting, orientation, and camera-specific characteristics. Hence, we can show the average reflectance of the leaves belonging to each class, hoping this information is enough to satisfy the reviewer’s request. This data is shown in the new Figure 1 and was obtained as follows (similar procedure to the one described in Section 2.5.1 to calculate the spectral variability of a sample leaf, graphically depicted in the first block of Fig. 3):
- Considering a certain leaf of class c, we load the normalized hypercube into memory and extract the spectra of each pixel, producing a matrix of size Npx x 120.
- According to the class, we concatenate the matrix obtained in (1), producing at the end a big matrix of size Npx*67 x 120 that contains all the individual matrices of the leaves belonging to class c.
- We compute the mean and the standard deviation over the columns of the matrix in (2), producing a vector of size 1 x 120 for the mean and another one of size 1 x 120 for the standard deviation (for each class c).
Remember that the ML models we trained use as input feature predictors, calculated from the VIs in Table 2 which are, in turn, calculated from the normalized reflectance contained in each pixel of the normalized hypercubes. We modified Section 2.2 to clarify the normalization procedure.
Q3: Figure 5. Move it to the results section.
A3: We wrote the article using LateX and, specifically, the MDPI Sensors template they made. As such, the software automatically places tables and figures where it feels it’s best for the overall quality of the layout. Figure 5 is coded inside the Results section, but it’s big (its size is a full page); hence, the software moves it where there’s space for it. We tried our best to place it in the main Results section nonetheless as requested.
Q4: Shorten the conclusion section. It is still too long.
A4: We shortened the conclusions section as requested. Removing more text will make this section too short in our opinion to be fully comprehended (also considering that most readers only read abstract and conclusions first, so they have to grasp the whole article from these two sections only).
Q5: In addition: Add real pictures of the samples and imaging setups.
A5: Real pictures of the leaves were already present in our previous article in ref. 30. We added them anyway as requested although it’s a duplication, in the new Figure 2. We don’t have a picture of the measurement setup to show.
Reviewer 3 Report
Comments and Suggestions for Authors
The esteemed authors have addressed all the concerns and issues I previously raised; no further comments are needed.
Author Response
We kindly thank the reviewer for his/her guidance.
Reviewer 4 Report
Comments and Suggestions for Authors
Authro provided careful and thorough revision. Author addressed the previous comments very well: the description of the data acquisition (two campaigns), the double Monte Carlo procedure, and the construction of the uncertainty-aware models and Bayesian validation are now clearly and systematically presented. The figures and tables effectively illustrate how incorporating uncertainty improves the detection of asymptomatic samples, and the link to your previous work is now well justified and well framed as a technical extension.
At this stage, I do not have additional scientific or methodological requests. I only recommend a light language and formatting polish before publication (e.g., correcting a few minor typos, smoothing a few long sentences, and checking consistency in reference ranges and notation for class labels and thresholds).
Author Response
We thank the reviewer for his/her kind guidance. We polished once again the revised manuscript and corrected all the minor issues we found.
Reviewer 6 Report
Comments and Suggestions for Authors
Dear authors, thank you for considering my suggestions. The abstract is now more informative, including quantitative details such as accuracy values and the best-performing model. The introduction has been partially condensed and states the objectives and novelty more clearly, although it remains somewhat lengthy and could still be streamlined for better focus. Methodological clarifications were added, including justification for the spectral range, the choice of quartiles, and the number of Monte Carlo iterations, which improves transparency. The discussion section shows slight improvement in biological interpretation, particularly regarding anthocyanin variability, but overall it still focuses heavily on numerical comparisons rather than physiological meaning or practical implications for vineyard management. Figures and tables remain dense; Figure 5 is visually overloaded, and Table 2 lacks fully consistent mathematical notation and clear indication of units. Citation style and abbreviation usage are more consistent, though minor issues persist. However, some key recommendations were only partially addressed: performance metrics beyond overall accuracy, such as precision, recall, and F1-score, were not included, and the conclusion could be more concise and forward-looking.
Author Response
Q1: The discussion section shows slight improvement in biological interpretation, particularly regarding anthocyanin variability, but overall it still focuses heavily on numerical comparisons rather than physiological meaning or practical implications for vineyard management.
A1: We understand the concern of the reviewer. However, due to the limited size of our dataset (based on data collected from a single vineyard), drawing conclusions about physiological meaning feels premature. For example, it is still to be investigated how the pattern appears on the leaf and why it appears on specific areas (a topic that we will investigate in the future, as addressed in the Conclusions). This topic is closely related to the physiological response of the plants (e.g. starch accumulation). Moreover, the disease affects differently red and white grape varieties, so we still have to test the white varieties to draw strong conclusions about the disease as a whole. As for the practical implications for vineyard management, we can simply say that our approach may be used to craft multispectral devices tuned on specific bands and coupled with our pipeline to detect early signs of FD in the field, as already stated in the final paragraph of Section 3.2.
Q2: Figures and tables remain dense; Figure 5 is visually overloaded, and Table 2 lacks fully consistent mathematical notation and clear indication of units.
A2: We understand that Fig. 5 appears imposing; however, we still think that splitting it (e.g. keeping the confusion matrices in a separate figure than the plots) will reduce understandability of our conclusions since they must be read at the same time. As for Table 2, all VIs are unitless since they are calculated as ratios of spectral reflectances which are unitless by definition. We added a short sentence about this in the caption of Table 2 and in the text of Section 2.3.2. Moreover, the mathematical notation adopted to describe the formulas is noted in the caption as well (single lambda means a reflectance value taken at a specific wavelength, while the new “overline” symbol means that multiple wavelengths were considered and averaged). If the reviewer has a specific mathematical notation in mind that he/she wish to see adopted in Table 2, we kindly ask to provide it or provide examples so we can better understand the request.
Q3: Citation style and abbreviation usage are more consistent, though minor issues persist.
A3: We wrote the article using LateX template of MDPI Sensors, and the bibliography was compiled using Bibtex. We checked the DOIs of all articles and corrected all the mistakes we found. Minor issues related to the citation style of the references will be addressed in post-production by the Editorial staff.
Q4: However, some key recommendations were only partially addressed: performance metrics beyond overall accuracy, such as precision, recall, and F1-score, were not included, and the conclusion could be more concise and forward-looking.
A4: Actually, performance metrics were already included as requested during the previous round of revisions in the new Table 4 (right before Fig. 5) and they were mentioned at the start of the Results section. Please take a look at page 17 of the latest revised article submitted.